

# The isotopic signatures of nitrous oxide produced by eukaryotic and prokaryotic phototrophs

Maxence Plouviez[1#], Peter Sperlich[2#] Benoit Guieysse[3], Tim Clough[4], Rahul Peethambaran[2], Naomi Wells[4]

[1]Cawthron Institute, Nelson, 7010, Aotearoa New Zealand
[2]National Institute of Water and Atmospheric Research (NIWA), Wellington, Aotearoa New Zealand
[3]BG Bioprocess Consulting, Palmerston North 4410, New Zealand
[4]Lincoln University, Lincoln, 7647, Aotearoa New Zealand

# These authors contributed equal amounts

*Correspondence to*: Naomi.Wells@lincoln.ac.nz

**Abstract.** Prokaryotic and eukaryotic microscopic phototrophs ('microalgae') can synthesize the potent greenhouse gas and ozone depleting pollutant nitrous oxide ($N_2O$). However, we do not know how much microalgae contribute to aquatic $N_2O$ emissions because these organisms co-occur with prolific $N_2O$ producers like denitrifying and nitrifying bacteria. Here we

demonstrate for the first time that microalgae produce distinct $N_2O$ isotopic signatures that will enable us to fill this knowledge gap. The eukaryotes *Chlamydomonas reinhardtii* and *Chlorella vulgaris*, and the prokaryote *Microcystis aeruginosa* synthesized $N_2O$ 265 – 755 $nmol \cdot g\text{-}DW^{-1} \cdot h^{-1}$ when in darkness and supplied with 10 mM nitrite ($NO_2^-$). The $N_2O$ isotopic composition ($\delta^{15}N$, $\delta^{18}O$, and site preference, SP) of each species was determined using a modified off-axis integrated-cavity-output spectroscopy analyser with an offline sample purification and homogenisation system. The SP values differed between

eukaryotic and prokaryotic algae ($25.8 \pm 0.3$ ‰ and $24.1 \pm 0.2$ ‰ for *C. reinhardtii* and *C. vulgaris*, respectively vs $2.1 \pm 3.0$ ‰ for *M. aeruginosa*), as did bulk isotope values. Both values differ from SP produced by denitrifiers. This first characterization of the $N_2O$ isotopic fingerprints of microscopic phototrophs suggests that SP-$N_2O$ could be used to untangle algal, bacterial, and fungal $N_2O$ production pathways. As the presence of microalgae could influence $N_2O$ dynamics in aquatic ecosystems, field monitoring is also needed to establish the occurrence and significance of microalgal $N_2O$ synthesis under

relevant conditions.

Keywords – Microalgae, Cyanobacteria, Eutrophication, Nitrous oxide, Laser-based spectroscopy



## 1 Introduction

Nitrous oxide ($N_2O$) is a strong atmospheric pollutant and one of the three major greenhouse gases with carbon dioxide ($CO_2$) and methane ($CH_4$) (Ciais, 2013; Tian et al., 2016; Tian et al., 2020). $N_2O$ is an intermediate molecule that is readily produced (and consumed) by a wide range of chemical and biological processes (Plouviez et al., 2018; Tian et al., 2020). For years, bacterial nitrification and denitrification were the only known major biological sources of $N_2O$ in the environment. However, it is now recognized $N_2O$ can also be emitted during fungal heterotrophic denitrification, archaeal ammonium oxidation and,

as most recently evidenced, microalgal $NO_3^-$ assimilation (Bellido-Pedraza et al., 2020; Plouviez et al., 2018; Teuma et al., 2023; Zhang et al., 2023). The ability of microalgae to synthesize $N_2O$ now challenges the 'bacteria-centric' view that all $N_2O$ emissions from aquatic ecosystems are related to bacterial metabolism (Plouviez and Guieysse, 2020; Plouviez et al., 2018; Teuma et al., 2023).

Under the Paris agreement, many countries have set stringent targets to reduce all greenhouse gases to net zero by 2050 (Den Elzen et al., 2025). This means that all sources need to be accounted for and that accurate methodologies are used for budgeting. In the case of $N_2O$, the Intergovernmental Panel on Climate Change provides guidelines that are based on nitrogen inputs, and where a proportion of these inputs is assumed to generate reactive nitrogen (e.g. ammonia, nitrate etc.) that can potentially form $N_2O$ (Webb et al., 2019; Teuma et al., 2023). While pragmatic, the IPCC method was shown to significantly

underestimate or overestimate $N_2O$ emissions from many aquatic ecosystems (Webb et al., 2019; Sun et al., 2025). Unpredictable aquatic $N_2O$ emissions are unsurprising given that $N_2O$ is a reactive intermediate species of multiple redox-regulated reactions and metabolic pathways (Stein and Klotz, 2016). Photoautotrophic $N_2O$ production further complicates this picture because these organisms influence oxygen availability, a parameter widely recognised to regulate $N_2O$ production vs consumption (Plouviez and Guieysse, 2020; Chang et al., 2022; Gruber et al., 2022). Process-specific monitoring is therefore

required for accurate inventories and mitigation strategies. Isotopic information on the processes driving $N_2O$ fluxes (Denk et al., 2017; Mccue et al., 2019) potentially provides an effective tool for improving the accuracy of greenhouse gas inventories (Park et al., 2012) and more efficient mitigation strategies (Gruber et al., 2022).

Small variations in the natural abundance of atoms with different mass of the same element (stable isotope signatures) have

been widely used to track interactions between, e.g., living organisms or waters (Glibert et al., 2018; Klaus and Mcdonnell, 2013). Stable isotopes are also a powerful tool to trace biogeochemical reactions – including identifying the source of greenhouse gases. This is because the biological and chemical processes that produce greenhouse gasses generally have a distinct 'preference' for light vs heavy isotopes that, once known, can be used to 'fingerprint' the origin of a given gas. For instance, isotopic signatures in methane produced from phytoplankton were used to verify the methane as biogenic (Klintzsch

et al., 2023). Isotope tracers can be particularly powerful for $N_2O$, where biogeochemical source information is imprinted on both its two stable isotopes ($\delta^{15}N$, $\delta^{18}O$) and the intermolecular position of $^{15}N$ within the molecule (site preference, SP-$N_2O$)

(Denk et al., 2017; Ostrom and Ostrom, 2017; Yu et al., 2020). SP-$N_2O$ is uniquely powerful because these signatures tend to be mass-independent, meaning that they do not vary with the reaction rate or the isotopic composition of the substrate. Yet these analyses require highly specialised and expensive equipment which has limited their development and implementation. Advances in laser technology promised affordable, high-throughput $N_2O$ isotopic analyses, but their environmental application remains limited by complex analytical effects due to sample matrix and non-linear instrument responses (Harris et al., 2020).

Here we describe a new method for the accurate laser-based analysis of $N_2O$ isotopes, which has enabled us to, for the first time, measure the SP-$N_2O$ signatures of microalgae. The study was therefore aimed to determine the SP-$N_2O$ signatures of $N_2O$ produced by microalgae as a first step to ultimately develop process-specific $N_2O$ monitoring from aquatic ecosystems.

## 2 Results and Discussion

### 2.1 $N_2O$ synthesis from *C. vulgaris*, *C. reinhardtii*, and *M. aeruginosa*

*C. vulgaris*, *C. reinhardtii*, and *M. aeruginosa* have been reported to synthesize $N_2O$ (**Table 1**). Following the protocol from (Plouviez et al., 2017), pure cultures of these three species were incubated in darkness and supplied with $NO_2^-$ to trigger $N_2O$ synthesis. The rates measured during this study are in the same order of magnitude to the ones reported previously for phototrophs (**Table 1**), however, lower than the rates reported by denitrifiers cultures ($2544 \pm 156$ nmol $N_2O \cdot h^{-1} \cdot g\text{-}DW^{-1}$, n = 3, further details about the denitrifier cultures can be found in Supplementary Information 1).

Microalgal $N_2O$ synthesis involves the reduction of $NO_2^-$ into nitric oxide (NO) and the subsequent reduction of NO into $N_2O$. In *C. reinhardtii*, $NO_2^-$ reduction into NO, is catalyzed by the dual enzyme nitrate reductase–NO-forming nitrite reductase (Plouviez et al., 2017) or the copper-containing nitrite reductase (Bellido-Pedraza et al., 2020). In darkness, NO reduction into $N_2O$, is then be catalyzed by cytochrome P450 (Plouviez et al., 2017; Burlacot et al., 2020). The presence of homologous proteins in *Chlorella vulgaris* and *Chlamydomonas reinhardtii* (Bellido-Pedraza et al., 2020) and biochemical evidence from (Guieysse et al., 2013) strongly suggest that *C. vulgaris* and *C. reinhardtii* synthesize $N_2O$ using a similar biochemical pathway. While a similar biochemical pathway was suggested by (Fabisik et al., 2023) for *M. aeruginosa*, this remains to be elucidated.

### 2.2 Performance of the modified off-axis integrated-cavity-output spectroscopy analyser after sample preparation on an offline matrix purification and homogenisation system

Our analytical approach (Sec. 4) accounts for complex challenges previously reported for this instrument type (Harris et al. 2020). Accuracy and reproducibility of the combined sample purification and laser analysis procedure was verified in each





measurement sequence by repeated analysis of a quality control standard.  For that purpose, aliquots of USGS52-in-air were

decanted in a sampling bag to be extracted, processed and analysed in the same way as the microalgae and cyanobacterial samples. This resulted in eight  USGS52-in-air measurements that we used to quantify the reproducibility of SP-$N_2O$ of 0.4 ‰ as per (Werner and Brand, 2001)  (**Figure 1**) and the accuracy of  –0.3 ‰, which is in agreement with the certified USGS52 value within the measurement uncertainty.

**2.3 SP-$N_2O$ values from *C. vulgaris*, *C. reinhardtii*, and *M. aeruginosa***

The eukaryotic microalgae (*C. reinhardtii* and *C. vulgaris*) and the cyanobacteria tested synthesized $N_2O$ and consistently produce a SP-$N_2O$ signature meaning there is a clear isotope preference during $N_2O$ production (**Table 1**). The SP-$N_2O$ signatures  of  the  eukaryotic  microalgae  were  similar  (25.8 ± 0.59 ‰  and  24.1 ± 0.37 ‰,  respectively)  and  significantly

different to the SP-$N_2O$ from *M. aeruginosa* (2.1 ± 6.8 ‰), meaning this could indeed be used to distinguish between photosynthetic $N_2O$ producers. The similarity of the isotopic signatures from the eukaryotic microalgae could be expected considering that both are chlorophyta, confirming that this division uses a consistent $N_2O$ biosynthetic pathway (Bellido-Pedraza et al., 2020; Plouviez et al., 2017). This needs to be further confirmed by testing other chlorophyta and eukaryotic taxons (Timilsina et al., 2022).


Overall, the values reported in this study are in systematic agreement with SP-$N_2O$ results from different categories of $N_2O$ sources. For the eukaryotic microalgae these signatures were distinct from bacterial denitrifiers (**Figure 2**). In contrast, cyanobacteria SP-$N_2O$ overlapped with bacterial denitrifiers. These findings suggest, first, that $N_2O$ isotopomer data from eutrophic waterways where both cyanobacteria and denitrifiers are likely to be abundant should be interpreted with care, and,

second, that SP-$N_2O$ could be used to untangle microalgal and denitrifier contributions to aquatic $N_2O$ emissions by comparing environmental signatures to site-specific SP-$N_2O$ community end-members. Using the full suite of isotopic information within the $N_2O$ molecule could greatly strengthen environmental identification of algal $N_2O$ production (see, e.g., (Wu et al., 2019)): the process potentially uniquely combine 'intermediate' SP values with isotopically depleted $\delta^{15}N$ (which even weak kinetic fractionation during $NO_2^-$ reduction to $N_2O$ would produce given the low yields, (Martin and Casciotti, 2016) and enriched

$\delta^{18}O$ (oxygen isotope effects are more complex, but high values could reflect oxygen exchange and associated equilibrium fractionation (Rohe et al., 2017; Barford et al., 2017).

**3 Environmental implications**

Nitrous oxide can be both produced and consumed by organisms (bacteria, fungi, archaea, plants – and algae) with very different life cycles, functions, and growth requirements. These organisms can synthesise $N_2O$ as an intermediate, by-product,

or end-product (Plouviez et al., 2018; Stein and Klotz, 2016; Bakken and Frostegård, 2017; Shan et al., 2021; Butterbach-Bahl

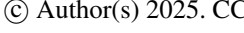

et al., 2013), which makes N$_2$O emissions particularly difficult to track when simply measuring changes in N$_2$O concentration. Empirical reports of contradictory responses to environmental fluctuations highlight the need for process-based measurements in order to accurately predict, and thus manage, aquatic N$_2$O emissions. Algal N$_2$O production could contribute to these seemingly contradictory responses. Preliminary testing of a mixed gas sample from a denitrification reactor and *C. vulgaris*

cultures showed that our methodology could provide the tool needed to start untangling this contribution as it will enable identify and quantify the source of the N$_2$O based on isotopic signatures in a mixed sample (Supplementary information 2). Further testing using environmental samples is now needed to establish the suitability of the method for N$_2$O process-specific monitoring.

Microalgae, including the ones selected for this study, are ubiquitous in the environment (Fabisik et al., 2023; Hou et al., 2023; Sasso et al., 2018). While we know that microalgae can synthesise N$_2$O, we actually know little about the occurrence and environmental significance of microalgal N$_2$O synthesis in ecosystems where algae are abundant, such as eutrophic environments (Plouviez et al., 2018). Human-related pollution causes massive eutrophication worldwide (e.g. 30 – 40% of the world's lakes are affected by eutrophication) so even a relatively 'modest' microalgal N$_2$O production could be globally

significant (Delsontro et al., 2019). Indeed, our findings suggest that some of the fluctuations in SP-N$_2$O reported during algae blooms (Glibert et al., 2018; Wang et al., 2023) could be due to N$_2$O production by microalgae. Understanding N$_2$O emissions from microalgae in aquatic ecosystems have, therefore environmental (climate science and nutrient management), and ecological (role of microalgae in N cycling) implications.

**Conclusions**

For the first time we characterized the isotopomer signatures of the microalgae *C. reinhardtii*, *C. vulgaris* and the cyanobacteria *M. aeruginosa*. This demonstrate that these phototrophs have the exhibit a clear signatures in isotopomers during N$_2$O production.

Importantly, the method should now be implemented for field samples to determine the true significance and dynamics of N$_2$O synthesis by microalgae in aquatic ecosystems.


**4 Appendix: Materials and Methods**

**4.1 Strain and culture maintenance**

*Axenic Chlamydomonas reinhardtii* 6145 was obtained from the Chlamydomonas resource center ([Home - Chlamydomonas Resource Center (chlamycollection.org)](https://chlamycollection.org)). Axenic *Chlorella vulgaris* UTEX 259 and *Microcystis aeruginosa* UTEX 2385 were

both obtained from the culture collection of the University of Texas at Austin ([https://utex.org/](https://utex.org/)). Pure cultures were maintained



on 250 mL TAP (*C. reinhardtii*, (Plouviez et al., 2017), BG 11 (*C. vulgaris*, (Guieysse et al., 2013)) and low-phosphate minimal media (*M. aeruginosa*, (Cliff et al., 2023)) incubated at 25ºC (INFORS HT Multitron) under continuous illumination (20 $\mu mol \cdot cm^{-2} \cdot s^{-1}$) with agitation (150 rotation per minutes, rpm) at a temperature of 25°C and a $CO_2$ supply (1% vol/vol). Cultures thus incubated for 2 weeks were re-suspended on fresh media 50% vol:vol.

**4.2 Cultivation and Bioassays**

The three species were grown as described above for 7 days. Following the protocol described by Guieysse et al. (2013), on the day of the experiment, 15 mL aliquots were withdrawn from the cultures to measure the cell dry weight (DW). Then, 25 mL aliquots were centrifuged at 4400 rpm for 3.5 min. The supernatants were discarded, and the pellets were re-suspended in N-free medium. Twenty-five mL aliquots of these suspensions were transferred into 120 mL serum flasks and supplied 10 mM
$NaNO_2$. The flasks were immediately sealed with rubber septa and aluminium caps and incubated at 25°C under continuous agitation (150 rpm) and darkness for 72 hours. Unless otherwise stated, cultures were run in triplicates. All glassware and media were autoclaved prior to the experiments.

**4.3 GC Analysis**

Gas samples (5 mL) were withdrawn from the flask headspace using a syringe equipped with a needle. The headspace $N_2O$
concentration in those samples was then quantified using gas chromatography (Shimadzu GC-2010, Shimadzu, Japan) as described by (Fabisik et al., 2023).

**4.4 Gas collection and bag preparation**

In line with (Gruber et al., 2022; Ding et al., 2025), we use aluminium-lined multi-layer foil gas sampling bags (3 L, Restek, https://www.restek.com/global/en/p/22950). Sample bags were flushed 3 times with instrument grade $N_2$. Each bag was then
filled with 1 L of instrument grade $N_2$. $N_2O$ gas sample withdrawn from the flasks` headspace were injected in the bag via the septa at the valve using a syringe and needle. Each bag had a final $N_2O$ concentration between 8 ppm and 16 ppm. The volume injected in each bag was specific to each flask and based on the $N_2O$ amount measured by the GC (ranging from 2 – 80 mL for the denitrifier and the eukaryotic microalgal cultures, respectively), before the samples were couriered to the National Institute of Water and Atmospheric Research (NIWA) facility in Wellington for SP-$N_2O$ analysis.

**4.5 Cryogenic Extraction of $N_2O$ from gas Samples for Isotopic Analysis**

A vacuum extraction line was built at NIWA to prepare the samples for SP-$N_2O$ analysis. This was needed to i) transfer the $N_2O$ into a natural air matrix to avoid air matrix artefacts, ii) to remove $H_2O$ and $CO_2$ as both species interfere with SP-$N_2O$





measurements in the analyser, and iii) to adjust the N$_2$O mole fraction to around 1 ppm to minimise N$_2$O amount effects (Harris
et al., 2020). A mass flow controller (0-300 mL/min, Bronkhorst, The Netherlands) was used to control the flow rate of the
sample gas and the N$_2$O-free air (**Table 2**). A first chemical trap containing magnesium perchlorate (Thermo Scientific, USA)
and Ascarite (Sigma Aldrich, USA) was used to removed H$_2$O and CO$_2$ (**Figure 3**).This is followed by two cryogenic traps
made from double loops of stainless-steel tubing with outer diameters of 1/2" (T1, large extraction trap) and 1/8" (T2, small
focus trap) that could be submerged in liquid nitrogen (LN$_2$).


For each sample, the volume of N$_2$O required to achieve a mixing ratio of 1 ppm in a 2500 mL mixing volume was calculated
based on the measured sample bag mixing ratio. The gas sample was then extracted at a flow rate of 100 mL/min until the
required sample amount was processed (solid blue arrows). This facilitated the trapping of N$_2$O molecules in the traps T1 and
subsequently in T2, both of which were maintained in LN$_2$. Subsequently, the volume of N$_2$O-free air required to achieve a 1
ppm concentration was passed through traps T1 and T2 at room temperature at a flow rate of 300 mL/min for 8 minutes and
20 seconds, respectively, carrying the extracted N$_2$O sample into the target bag (dashed black arrows).

### 4.6 Isotopic analysis of SP-N$_2$O

4.6.1. SP-N$_2$O analyser and considerations of known analytical challenges
Site preference in N$_2$O (SP-N$_2$O) was measured using an optical analyser (model N$_2$OIA-23e-EP , Los Gatos Research, USA),
referred to as LGR throughout. This continuous-flow analyser operates at sample flow rates of 80 mL/min, it has an optical
cavity volume of ~900 mL and operates at gas pressures around 57 mbar within the cavity. The LGR responds to pressure
changes at the sample inlet port by gradually adjusting the cavity pressure with a time lag, causing a pressure-dependent bias
in the reported SP-N$_2$O and N$_2$O mole fractions (Radu et al., 1998). Moreover, this instrument includes a significant N$_2$O
concentration bias, where reported SP-N$_2$O values can vary strongly with N$_2$O mole fractions, following a non-linear function
(Griffith, 2018; Harris et al., 2020). Consequently, differences in gas pressure within the cavity, the presence and amount of
interferant gases and in the N$_2$O mole fractions between the measurements of samples and reference gases, need to be carefully
controlled and accounted for to achieve accurate SP-N$_2$O measurements of the samples (Harris et al., 2020). The following
sub-sections describe the required steps to achieve accurate and reproducible SP-N$_2$O measurements using the LGR.


4.6.2. Control of cavity pressure and interferants: Modified gas inlet and sample control system
To achieve accurate SP-N$_2$O measurements a modified sample inlet system was installed inside the LGR (**Figure 4**). This
modification allowed changing of the LGR operation from continuous flow mode to a discrete mode by switching gas flows
using solenoid valves (Series 9 and Series 99, Parker, USA). In discrete mode, the flow scheme includes a cylindrical stainless
steel volume of 30 mL, which we refer to as the Mixing Volume (MV) (**Figure 4**). Four solenoid valves are welded onto the
MV to: i) inject sample gases and N$_2$O-free air into the MV, ii) to inject the sample into the cavity of the LGR and iii) to
connect a vacuum pump (model XDS 35i, Edwards, UK) for evacuation. This pump is also used to evacuate the analyser cavity



to ~0.02 mbar between samples via a solenoid valve with a large diameter orifice to ensure rapid evacuation (3/8" orifice, A15 type, Parker, USA) installed at the cavity outlet. The MV was furthermore equipped with a pressure gauge (0-2.5 bar, 21Y model, Keller Pressure, Winterthur, Switzerland). A chemical trap with magnesium perchlorate (Thermo Scientific, USA) and Ascarite (Sigma Aldrich USA) is installed upstream of the sample gas port to remove both $H_2O$ and $CO_2$, respectively, from samples (and reference gases) to below 5 ppm $H_2O$ and 0.5 ppm $CO_2$ (Sperlich et al., 2022). A manifold of eight solenoid valves (V100, SMC, Japan) allowed injecting $N_2O$-free air, two SP-$N_2O$ reference gases, one quality control standard and up to four samples through the scrubber into the MV (**Figure 4**). The sample inlet system is fully automated through a LabView interface (National Instruments, Austin, Texas, USA), combining gas control through scripted measurement sequences and the acquisition of all LGR and gas control data within a single output file. With this system, sample and reference gases can be injected into the MV at controlled pressures, achieving an average variation of 0.5 ± 0.3 mbar (1 σ), resulting in an average magnitude of 0.6 ± 0.7 ‰ (1 σ) for the pressure correction of reported SP-$N_2O$ values.

### 4.6.3. Gases used

Gases used for sample preparation or as standards are summarised in **Table 2**. At the time of publication, reference gases for $N_2O$ mole fractions covering an upper range of 3.5 ppm were not available to our laboratory. Therefore, $N_2O$ mole fractions in this study are only used for sample processing purposes and are otherwise treated as "indicative". A working standard (1080) was prepared by filling a cylinder with clean, Southern Ocean baseline air with the addition of pure $N_2O$ to achieve a mole fraction of around 1.080 ppm. Blocks of this working standard were implemented into each measurement sequence to monitor and correct for instrumental drift. We used "cryogenically purified air" (Praxair, California USA) with a certified $N_2O$ mole fraction blank of <1 ± 1 ppb as $N_2O$-free air. $N_2O$-free air is used to flush the analyser as well as for the dilution of sample and reference gases.

The instrument was calibrated for SP-$N_2O$ using USGS51 and USGS52 (Ostrom et al., 2018), purchased from the US Geological Survey and with isotope values shown in **Table 3**. Aliquots of both gases were transferred into 30 L Luxfer cylinders (Praxair, California USA) and diluted with $N_2O$-free air to target mole fractions of 3.5 ppm. This resulted in two cylinders, one each with USGS51-in-air and USGS52-in-air mixtures at filling pressures of 40 bar, for which we applied the isotope values (**Table 3**) of the USGS51 and USGS52 certification (Ostrom et al., 2018).

### 4.6.4. Matching $N_2O$ amounts in samples and reference gas to minimise $N_2O$ amount correction

Following the extraction, purification and dilution, all samples were connected to the sample inlets on the LGR and tested for their indicative $N_2O$ mole fraction first. Indicative $N_2O$ mole fraction values were used to calculate the dilution factor needed to match $N_2O$ mole fractions between each sample and each bracketing reference gas measurement. These dilution factors are incorporated into each measurement sequence to control valve switching times and target pressures during the injection of samples, reference gases and $N_2O$-free air into the MV. For example, the target $N_2O$ mole fraction in the extracted samples





was 1 ppm. With a $N_2O$ mole fraction of 3.5 ppm in the USGS51-in-air reference gas, the latter needed to be diluted with $N_2O$-free air to match the mole fraction of the sample of 1 ppm. The system achieved average $N_2O$ mole fraction matches within $61 \pm 42$ ppb (1 σ), resulting in an average $N_2O$ correction concentration of $1.4 \pm 0.9$ ‰ (1 σ). While this strategy is technically

cumbersome, it minimises the uncertainty of applying $N_2O$ amount corrections, which is non-linear, time-variable and found to have a magnitude between 5 ‰ and 25 ‰ when the mole fraction ranges from 0.45 ppm to 1.5 ppm. **Figure 5** shows the SP-$N_2O$ values determined within each measurement sequence when determining the $N_2O$ amount effect. While the day to variability can be very small, this artefact shows considerable variability with time and therefore needs regular quantifying.

4.6.5. Controlling $N_2O$ mole fraction bias and instrumental drift in measurement sequence and protocol

A schematic overview of the measurement sequences is shown in **Figure 6**. Measurement sequences comprise of a series of measurement blocks of SP-$N_2O$ reference gases, working standards and samples. Each gas sample was injected into the pre-evacuated cell of the LGR, before being locked in and measured for ten minutes before the cell was evacuated and flushed with $N_2O$-free air in preparation for the consecutive analysis. Up to five blocks of the working standard (1080) are measured

within each sequence to monitor instrumental drift. Following the first 1080 block, the $N_2O$ amount correction function is determined. For that, reference gases are diluted with $N_2O$-free air inside the MV without changing their SP-$N_2O$ prior to their injection into the LGR. USGS51-in-air and USGS52-in-air are each analysed at five $N_2O$ mole fraction levels over a range of 0.33 to 1.5 ppm with five repetitions per $N_2O$ level. This is followed by measurements of up to four samples, each of which is bracketed by blocks of ten USGS51-in-air measurements at matching $N_2O$ mole fractions. The pathway to inject samples and

reference gases includes further purification with a chemical scrubber and all steps of gas handling and analysis follow the principle of identical treatment (PIT) (Werner and Brand, 2001) as much as possible.

4.6.6. Data processing

The LabView interface generates a data file including all raw data from the LGR as well as sample handling data and instrument

performance data from the sample inlet with a time resolution of 1 Hz. Data processing starts with data reduction to generate averages for all output data. Next, all data are corrected for variation in cell pressure, following an experimentally determined, linear correction function. Thereafter, the pressure-corrected measurements of the $N_2O$ amount effect determination are assessed for analyser drift using the first three blocks of the 1080 working standard. A correction is only applied when the drift effect is significant and exceeds twice the measurement reproducibility in SP-$N_2O$ (~1 ‰). The next step normalises the SP-

$N_2O$ values of the samples relative to the bracketing USGS51-in-air measurements and applies the correction for $N_2O$ mole fraction differences. The final step applies the $N_2O$ mole fraction correction to the data from the $N_2O$ amount effect determination, resulting in fully corrected measurements of USGS51-in-air and USGS52-in-air, which are then used for a two-point calibration to the SP-$N_2O$ values of the samples. Because the range of $\delta^{15}N$ and $\delta^{18}O$ values covered by USGS51 and USGS52 is too small for a two-point calibration, we determined the $\delta^{15}N$-$N_2O$ and $\delta^{18}O$-$N_2O$ results from algae based on a

one-point calibration using USGS51-in-air.



4.6.7. Measurement reproducibility, accuracy and propagated uncertainty

For SP-N$_2$O in each sample, we derived the total uncertainty (U$_{tot}$) by propagating all contributing uncertainties as the square root of the sum of squares:

$U_{tot} = SQRT(U_{sam}^2 + U_{REF\_a}^2 + U_{REF\_b}^2 + U_{REF\_span}^2 + U_{p-corr}^2 + U_{N2O-amount-corr}^2)$, Equation 1

Where U$_{sam}$, U$_{REF\_a}$, U$_{REF\_b}$ and U$_{REF\_span}$ represent the standard deviations (1 σ) of the measurements of the samples, the two bracketing USGS51-in-air reference gases before (_a) and after (_b) the sample measurement, as well as the USGS52-in-air measurement used for the final isotope span correction for SP-N$_2$O, respectively. U$_{p-corr}$ and U$_{N2O-amount-corr}$ represent the uncertainties of the correction for gas pressure variation in the cell as well as the N$_2$O amount effect, which are calculated as

the standard error of the mean of the residuals of each correction function. Uncertainties of $\delta^{15}$N and $\delta^{18}$O values are calculated in the same way, except they do not include the uncertainty of USGS52-in-air as a two-point calibration is not applied.

**Data availability**

All the data from our study are presented numerically in the paper and in the Supplement.


**Supplement**

Supplementary Information 1: Denitrifiers cultures; Supplementary Information 2: Analytical blind test using isotope analysis to determine fractions of N$_2$O from two biological sources in a gas mixture.

**Author contributions**

MP and PS performed the investigation, data visualization and curation and were involved with the writing (original draft) and contributed to conceptualization, methodology and data curation, as well as visualization with NW, BG and TC. RP provided support with gas samples preparation and spectroscopy analyses. NW, BG, RP and TC were involved in the writing (review and editing) of the paper before submission. Finally, all authors except RP were involved with the funding acquisition.


**Competing interests**

The contact author has declared that none of the authors has any competing interests.

**Acknowledgements**

We acknowledge financial support through the MLAIT funding scheme between Massey and Lincoln Universities, as well as NIWA's Strategic Science Investment Funding through the Understanding Atmospheric Composition and Change programme. We gratefully acknowledge invaluable support and discussions during the development of the SP-N$_2$O instrument from Mike Harvey, Ross Martin, John McGregor and Colin Nankivell.



**Financial support**

Massey – Lincoln and Agricultural Industry Trust (46524) and NIWA's Strategic Science Investment Funding.

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



**Tables**

**Table 1**: The production rate and isotope composition of $N_2O$ by two eukaryotic microalgae (*C. reinhardtii* and *C. vulgaris*)
and one prokaryotic cyanobacteria (*M. aeruginosa*). Values are reported as the mean ± SD of laboratory replicates, and the range of analytical uncertainty of the individual SP measurement ($U_{tot}$) are also shown. Letters indicate differences between species.

| Species | n | $N_2O$ production rate (nmol $N_2O$/g DW/hr)[a] | $\delta^{15}N_{N2O}$ (‰ v AIR)[b] | $\delta^{18}O_{N2O}$ (‰ v VSMOW)[c] | SP-N2O [d] | $U_{tot}$ SP | Reference |
|---|---|---|---|---|---|---|---|
| *C. reinhardtii* | 4 | 370 ± 87 | -105 ± 0.73 | 26.4 ± 1.7 | 25.8 ± 0.59 | 1.1 – 1.2 | This study |
| | | 52 – 1,100 | - | - | - | - | (Plouviez et al., 2017; Burlacot et al., 2020; Bellido-Pedraza et al., 2022) |
| *C. vulgaris* | 5 | 740 ± 390 | -115 ± 0.45 | 29.7 ± 0.80 | 24.2 ± 0.37 | 0.8 – 1.2 | This study |
| | | 1,000 – 1,700 | | | | | (Guieysse et al., 2013) |
| *M. aeruginosa* | 5 | 510 ± 150 | -115 ± 3.6 | 11.4 ± 4.0 | 2.12 ± 6.8 | 1.0 – 1.7 | This study |
| | | 170 – 230 | | | | | (Fabisik et al., 2023) |

a F = 62, *p*<0.0001

b F = 1200, *p*<0.0001

c F = 41, *p*<0.0001

d F = 45, *p*<0.0001





**Table 2**: Information on gases used in this study.

| Gas name | Components | Origin | Indicative N₂O mole fraction | Functional use |
|---|---|---|---|---|
| USGS51-in-air | USGS51 + N$_2$O-free air | USGS (N$_2$O) Praxair (N$_2$O-free air) | 3.5 ppm | SP-N$_2$O calibration standard |
| USGS52-in-air | USGS51 + N$_2$O-free air | USGS (N$_2$O) Praxair (N$_2$O-free air) | 3.5 ppm | SP-N$_2$O calibration standard |
| 1080 | Natural air + pure N$_2$O | NIWA (natural air) BOC (pure N$_2$O) | 1.080 ppm | Working standard, instrument drift |
| N$_2$O-free air | Cryogenically purified natural air | Praxair (N$_2$O-free air) | N$_2$O-free | Gas dilution, instrument flushing |




**Table 3**: Certified SP-$N_2O$ values for USGS51 and USGS52, adopted for the USGS51-in-air and USGS52-in-air reference gases (Ostrom et al., 2018) and used for value assignment of the reported measurements in a 2-point calibration for SP-$N_2O$. Both $\delta^{15}$N-$N_2O$ and $\delta^{18}$O-$N_2O$ are determined in a 1-point calibration based on USGS51 only.

| Gas name | SP-$N_2O$ | $\delta^{15}$N-$N_2O$ | $\delta^{18}$O-$N_2O$ | 470 |
|:---:|:---:|:---:|:---:|:---:|
| USGS51 | −1.67 ‰ | +1.32 ± 0.04 ‰ | +41.23 ± 0.04 ‰ | |
| USGS52 | +26.15 ‰ | | | |



**Figures**

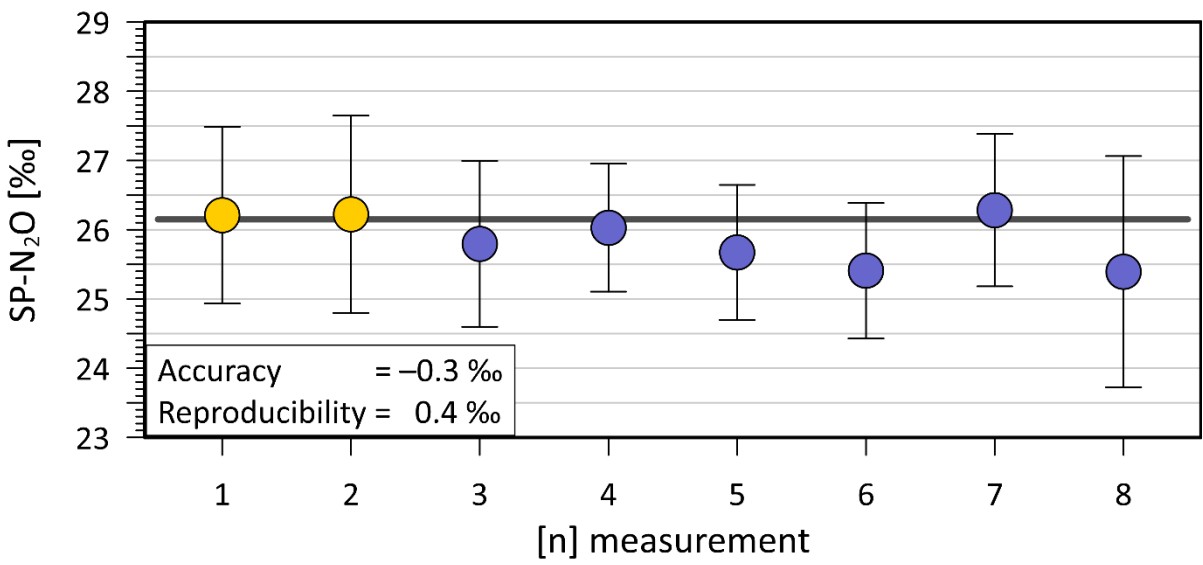


**Figure 1**: Reproducibility of the analytical system for SP-N$_2$O. Orange symbols show SP-N$_2$O results from USGS52-in-air measurements used to verify the robustness of the extraction system. Blue symbols show measurements of USGS52-in-air as quality control standard during the measurements of unknown samples. Error bars indicate the propagated uncertainty for each measurement. The thick black line indicates the target value of 26.15 ‰ for USGS52.




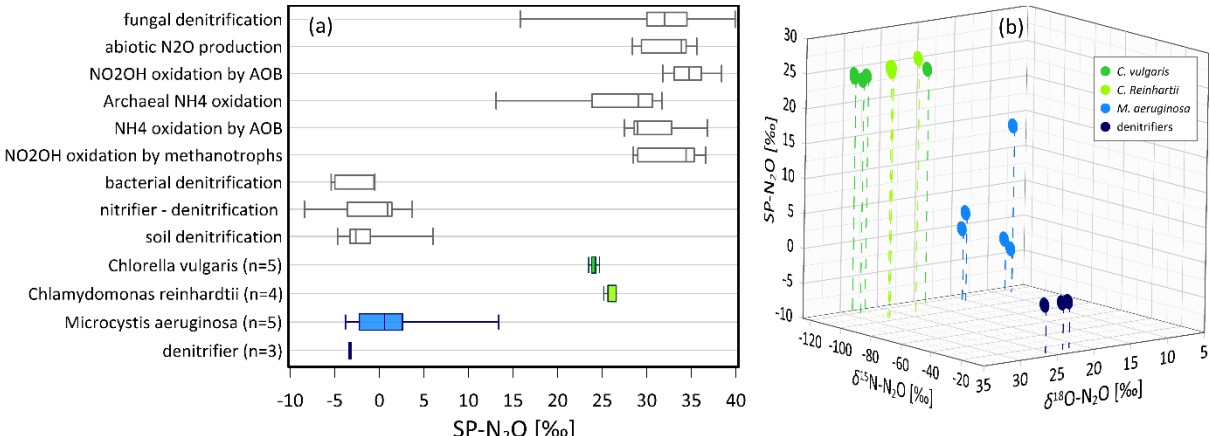

**Figure 2**: (a) The range of SP-N$_2$O reported for different N$_2$O production pathways from previously published values (Denk
et al., 2017) in white or values obtained from this study in colour. The centre lines of the box show the median, the box edges
the quartiles and the whiskers represent minimum/maximum values. (b) 3D N$_2$O isotopes plot for microalgal, cyanobacterial
and denitrifier samples (same colour as in the box-whisker plot). See **Table 1** for microalgae and cyanobacteria N$_2$O production
rates.



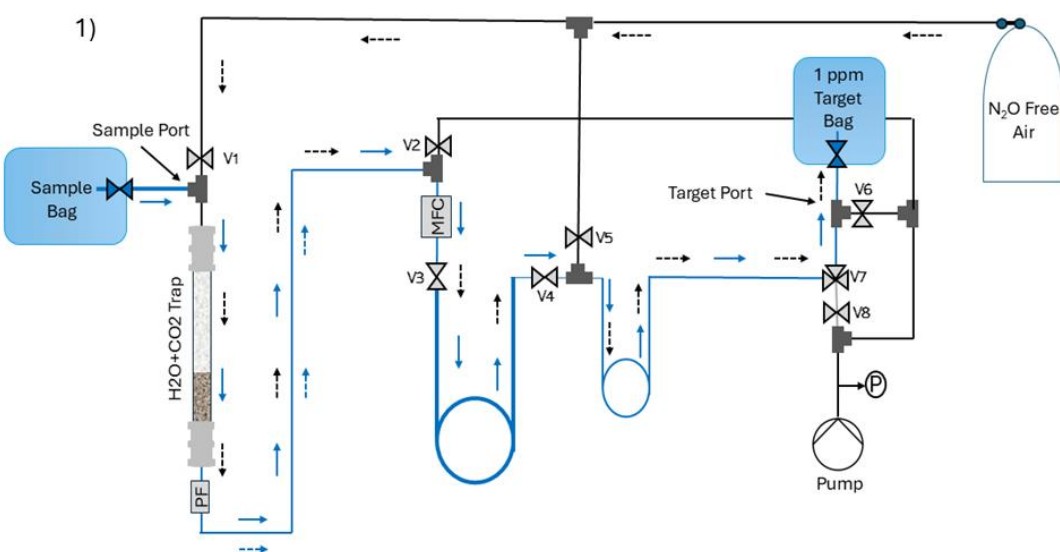

**Figure 3**: Schematic of the N₂O extraction line with sample and target bags. The flow pathways for the sample gas are indicated by solid blue arrows, while the flow of N₂O-free air is represented by black dashed arrows.



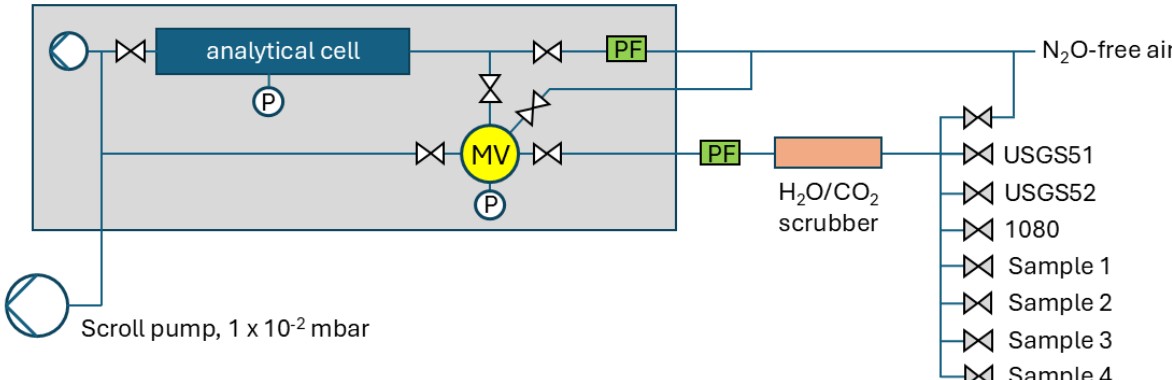

**Figure 4**: Schematic of the LGR $N_2O$ isotopomer analyser (grey box), showing the installed mixing volume with pressure
gauge and solenoid valves for gas handling and dilution. External additions include the scroll pump, as well as a particle filter
and a $CO_2$ scrubber to remove $H_2O$ and $CO_2$ in gas samples supplied from the valve manifold.





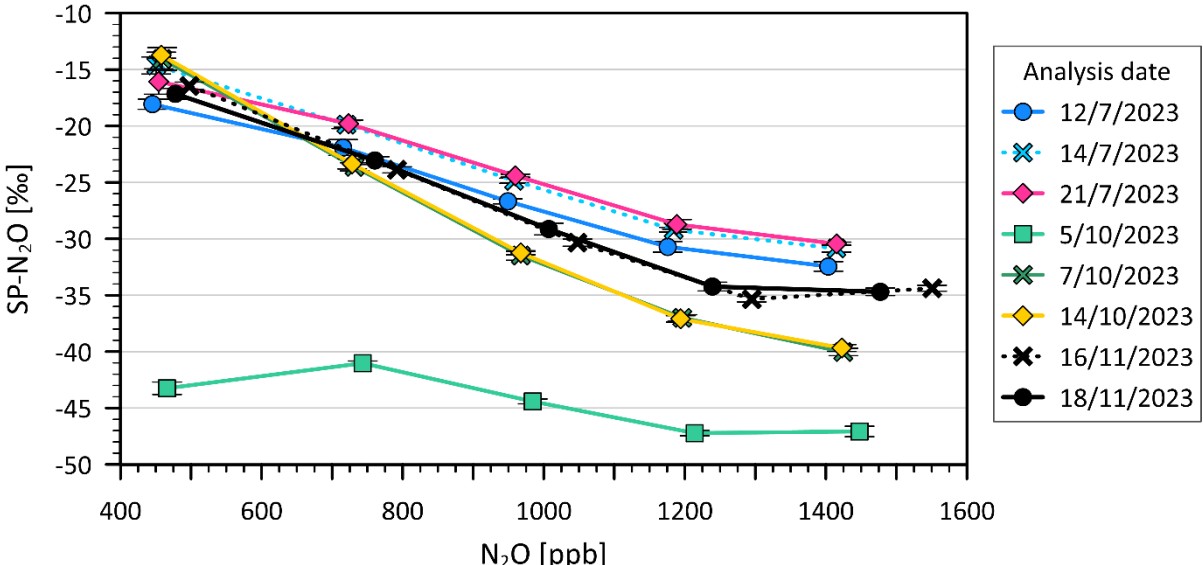

**Figure 5.** Day-to-day variation of measurement bias in SP-$N_2O$ due to $N_2O$ amount dependence as determined with USGS51-
in-air at the start of each measurement sequence. Error bars indicate the standard deviation (1 $\sigma$) of the SP-$N_2O$ measurements
at the respective $N_2O$ level.





| Block # | 1 | 2 | 3 | 4 | 5 | 6 | 7 | 8 | 9 | 10 | 11 | 12 | 13 | 14 | 15 | 16 | 17 | 18 | 19 |
|---|---|---|---|---|---|---|---|---|---|---|---|---|---|---|---|---|---|---|---|
| Gas | WT | USGS51-in-air | WT | USGS52-in-air | WT | USGS51-in-air | S1 | USGS51-in-air | USGS51-in-air | S2 | USGS51-in-air | WT | USGS51-in-air | S3 | USGS51-in-air | USGS51-in-air | S4 | USGS51-in-air | WT |
| N$_2$O amount | 1080 | 400-1500 | 1080 | 400-1500 | 1080 | match S1 | 1000 | match S1 | match S2 | 1000 | match S2 | 1080 | match S3 | 1000 | match S3 | match S4 | 1000 | match S4 | 1080 |
| purpose | drift | N$_2$O amount SP assignment | drift | N$_2$O amount SP assignment | drift | S1 drift correction | | S1 drift correction | S2 drift correction | | S2 drift correction | drift | S3 drift correction | | S3 drift correction | S4 drift correction | | S4 drift correction | drift |
| n repetitions | 5 | 5 x 5 | 5 | 5 x 5 | 5 | 10 | 10 | 10 | 10 | 10 | 10 | 5 | 10 | 10 | 10 | 10 | 10 | 10 | 5 |

**Figure 6**: The Measurement sequence combines blocks of measurements of air from the working tank (1080), the USGS51-in-air and USGS52-in-air standards used to define the correction for the N$_2$O amount effect as well as scaling for a two-point calibration of SP-N$_2$O values, up to four samples (S1-S4) and bracketing blocks of USGS51-in-air measurements with N$_2$O amounts that match the N$_2$O amount in their respective sample.