# Peer review of "The isotopic signatures of nitrous oxide produced by eukaryotic and prokaryotic phototrophs"

_EGUsphere, 2025_

## Referee Comment (RC2)

**Reviewer Comments on Plouviez et al. 2025**

**General comments**

Plouviez et al. take on a significant problem within the biogeosciences – namely, that the $N_2O$ budget is not closed, and one of the main hurdles in closing the budget is accurately accounting for all sources and sinks of $N_2O$ since it can be produced / consumed by multiple biotic and abiotic pathways. A powerful tool in this space is measuring "Site Preference" (SP), which quantifies the relative 'preference' of $^{15}N$ for the central ('alpha') or outer ('beta') site in the asymmetrical, linear $N_2O$ molecule. Therefore, many groups have been working to systematically measure the SP of all known sources and sinks of $N_2O$ in an effort to close the $N_2O$ budget, as well as identify sources of $N_2O$ that may be mitigated to prevent greenhouse gas emissions.

Plouviez et al. measure the $N_2O$ SP of eukaryotic and bacterial photosynthesizers, which have been shown to produce $N_2O$ outside of the metabolic pathways that $N_2O$ production has been typically attributed to (denitrification and nitrification, either by bacteria or fungi). Measuring this value is particularly important for understanding $N_2O$ cycling in the oceans, since denitrifiers, nitrifiers, and algae all coexist in complex microbial assemblages – therefore, finding potentially unique SP signatures for algae may help disentangle complex marine $N_2O$ cycling. They state that they describe a new method for the accurate laser-based analysis of $N_2O$ isotopes, which enables them to conduct novel SP measurements of algal $N_2O$. They find significantly different SP signatures for the eukaryotic algae (*C. reinhardtii* and *C. vulgaris*) compared to the cyanobacteria (*M. aeruginosa*).

I have two main comments regarding this manuscript. The first is about the technical aspects of the measurement. The second is about the interpretation of the isotopic signatures.

For the first point, I cannot independently evaluate the quality of the data presented because it lacks key outputs that would enable independent calculation of this. I assume that authors are measuring the major isotopologues $^{14}N^{14}N^{16}O$ (446), $^{14}N^{15}N^{16}O$ (456), $^{15}N^{14}N^{16}O$ (546) and $^{14}N^{14}N^{18}O$ (448) and not the rarer clumped species (i.e. $^{14}N^{15}N^{18}O$), though this is never explicitly stated. SP is calculated as the relative difference between the $^{15}N$ isotopologues (SP = $\delta456 - \delta546$) and the bulk nitrogen isotopic composition is the average of the alpha and beta sites ($\delta^{15}N_{bulk} = (\delta456 - \delta546)/2$; see Kanterova et al. 2022 *GCA*, for example, for calculations) – therefore, understanding issues of sample bracketing, variations among samples, and etc. can be masked by reporting SP only. This is because variations in SP can be driving by variations in one isotopologue alone, since SP simply describes a relative difference in 456 and 546. In addition, as noted in Griffith 2018 *GMT*, several commercial manufacturers offer optical analyzers based on laser or FTIR spectroscopy that report results in various ways – as an isotopologue mole fraction and/or total mole fractions and/or in 'traditional' isotope delta values. Plouviez et al. do not report the equations used to convert from raw, instrument measurements to final delta values. They also do not report $\delta456$ and $\delta546$ (also denoted as $\delta^{15}N$-alpha and $\delta^{15}N$-beta), nor do they show that the calculated $\delta^{15}N_{bulk}$ from these values match the measured $\delta^{15}N_{bulk}$ values. They do report some necessary data in reporting "a new method" – i.e. Figure 1 and 5 – but, again. they do not report their full dataset and only their final calculated values. For

example, Table 1 gives the averaged isotopic measurements across all replicates for each species, but the individual measurements behind each average are not in the main text or supplement. Therefore, it is difficult to independently evaluate the quality of their data. I would encourage the authors to publish a more complete dataset, as well as equations involved in converting from raw, instrument measurements to final, reported delta values. This could be amended to the existing supplemental.

For the second point, I would: 1) Encourage the authors to comment more on the potential mechanism behind the large difference in SP values between the eukaryotic vs. bacterial algal strains; and 2) Have some clarifying questions regarding controls in their experimental systems. As the authors are likely aware of, in both eukaryotic and bacterial algae, it is thought that there are primarily two sources of $N_2O$: flavodiiron proteins (FLV) and cytochrome p450s (CYP55). FLVs are used in pseudo-cyclic electron flow for Photosystem I (PSI) photoprotection, where electrons are put onto $O_2$ instead of being used to generate NADPH. It has been shown that NO can be reduced instead of $O_2$, generating $N_2O$ in the process (Burlacot et al 2020 *PNAS*). CYP55s are a broad class of enzymes involved in multiple metabolic pathways, including pigment biosynthesis and lipid metabolism – i.e., reactions not involved in the light reactions of photosynthesis. Hence, as noted by the authors, it has been shown in *C. reinhardtii* that FLV produce $N_2O$ in the light, while CYP55 produces $N_2O$ in the dark (Burlacot et al. 2020 *PNAS*). Due to similarities between the species, *C. vulgaris* should use a similar pathway, as noted by the authors. Prior work by some of the authors (Fabisik et al. 2023 *Biogeosciences*) performed a BLASTP search on *M. aeruginosa* and found hits for FLV and CYP55, suggesting that similar pathways exist in this strain as well.

Plouviez et al. perform all cell suspensions in the dark – this should isolate the CYP55 signal. The SP signals from *C. reinhardtii* and *C. vulgaris* are similar to that of the fungal nitric oxide reductase (Figure 2), which also belongs to the CYP55 family. However, the SP signals from *M. aeruginosa* are quite different and better match the bacterial nitric oxide reductase (Figure 2). One interpretation of their results is that CYP55 from eukaryotic and bacterial algae are quite different, and that is reflected in their $N_2O$ SP values – this appears to be the primary interpretation that the authors make, though they do not attribute it to the enzyme explicitly. Alternatively, in *M. aeruginosa*, since they note that the pathway has not been fully 'elucidated,' non-CYP55 sources of $N_2O$ may be possible. Potentially relevant, an enzyme called flavohemoglobin protein (FHP) has recently been measured for $N_2O$ SP (Wang et al. 2024 *PNAS*). FHP is similar to FLV as they both have flavins as a co-factor – diflavins like FLV have two, while FHP has a flavin and heme cofactor. Measured $N_2O$ SP values in Wang et al. 2024 *PNAS* of FHP from *P. aeruginosa*, *A. baumannii* and *S. aureus* are similar to those measured from *M. aeruginosa* in this paper (roughly 0 to 15‰ in Wang et al., depending on strain, compared to 2±7‰ for *M. aeruginosa* in this paper). The authors should also explicitly note if *M. aueruginosa* has a nitric oxide reductase (NOR) or not, since that would aid in interpretation of this unique signal. In addition, it may be potentially relevant that the standard deviation of their reported SP values from *M. aeruginosa* is much larger than that of *C. reinhardtii* and *C. vulgaris*, though they do not report the non-averaged data nor the 456 / 546 data, so this is difficult to interpret. That may also help better interpret why the SP values of *M. aeruginosa* are so different.

For this point, did the authors repeat their experiments in the light? Given the established light-dependent nature of $N_2O$ production in *C. reinhardtii*, if FLV does have a different $N_2O$ SP, one would expect to see a shift in the $N_2O$ SP of *C. reinhardtii*.

In addition, other enzymes besides NOR, FLV, and CYP55 can produce $N_2O$. Currently, nitric oxide reductases (NOR), P450nor, cytochrome P460, cytochrome p450 (CYP55), cytochrome c554, flavodiiron proteins (FLVs) and flavohemoglobin proteins (FHPs) have been shown to produce $N_2O$ as a direct product of an enzymatic reaction (see Ferousi et al. 2020 *Chem Rev*, Kuypers et al 2018 *Nat Rev Microbiol* and Poole & Hughes 2000 *Mol Microbiol* for review). Did the authors attempt to check if their wild-type (WT) strains have genes encoding any of these potential enzymatic sources? This check can be doing through searches like BLASTP, qPCR, RNAseq, or other similar techniques for working on non-genetically tractable strains (i.e. strains where making clean deletions of a certain gene are difficult).

Finally, regarding experimental controls, it is established that $N_2O$ can be produced abiotically (i.e. 'chemodenitrification' Stanton et al. 2018 *Geobiology*), and this process is strongly pH-dependent, where acidic pHs produce nitric oxide radicals that can then be further reduced to N2O (i.e. Su et al 2019 *ES&T*). Did the authors control or check pH of their growth media? Though the media composition is given, there is no indication that the pH of the system was checked prior to incubation, or what the target pH of their media is. In addition, did the authors perform any no-cell controls, where the media was incubated with no cells? I may have missed this, but it does not appear that the authors did this. In addition, $N_2O$ can be formed readily from NO radicals, which makes it important to control for all sources of NO radicals, particularly in wild-type (WT) strains. Both bacteria and eukaryotes can create NO through a diverse set of nitric oxide synthases (Forstermann & Sessa 2011 *Eur Heart J*), and these NO radicals can spontaneously react to form $N_2O$ in the absence of oxygen. Did the authors check for these potential NO sources in their strains?

Overall, Plouviez et al. tackle an important problem in the biogeosciences – constraining the $N_2O$ SP of eukaryotic and bacterial photosynthesizers, which produce $N_2O$ outside of the metabolic pathways that $N_2O$ production has been typically attributed to (denitrification and nitrification, either by bacteria or fungi). Their work offers an important starting point for further, more detailed physiological work that will enable this measurement to be used to disentangle complex microbial communities of denitrifiers, nitrifiers, and photoysnthesizers, helping the community close the $N_2O$ budget and disentangle complex marine N2O cycling.

**Specific questions**

The authors fine extremely depleted $\delta^{15}N_{bulk}$ values of about –100‰. This is outside of the range of their standards, and also of $N_2O$ in air (ranged from ~9 to 6‰ over the past 300 years; Park et al 2012 Nat Geosci). Did they use a very depleted source of nitrite? The $\delta^{15}N$ of the nitrite supplied should be included.

In Table 1, what does "F" mean in the footnotes? At first I thought it meant fraction consumed, but one value of F is 1200.

For Table 3, what are the $\delta^{15}N$-alpha and $\delta^{15}N$-beta values for the standards used?

For Figure 1, what are the $\delta^{15}$N-alpha and $\delta^{15}$N-beta values, not just the SP values? In addition, just to clarify, only USGS52 was measured over time, and not USGS51 as well? Related to this, I am slightly confused because Figure 5 shows only USGS51 and not USGS52, and Figure 6 suggests that both reference gases were measured regularly.

For Figure 2a, Wang et al. 2024 *PNAS* offers a more recent compilation of $N_2O$ SP measurements than Denk et al. 2017.

For Figure 2b, the authors are comparing their data vs. that from published denitfier data. In the text (line 113) and in the figure legend, which experimental denitrifer data are the authors comparing their data to? (In addition, the plot should specific 'bacterial denitrifiers' instead of just 'denitrifiers'). Multiple groups have measured bacterial denitrifiers and there is a larger range of values than they show in their figure. For example, see Wang et al. 2024 *PNAS* or Toyoda et al 2017 *Mass Spectrom Rev* for recent compilations. In addition, unless the authors are using nitrite with the exact same $\delta^{15}$N as that study, one would not expect the $\delta^{15}$N-$N_2O$ and $\delta^{18}$O-$N_2O$ to be the same. Instead, the relative fractionation (i.e. $^{15}\varepsilon$ or $^{18}\varepsilon$) are comparable, not the bulk values. Therefore, the epsilons should be calculated and plotted instead.

For Figure 5, this is something where showing the full suite of data ($\delta^{15}$N-alpha, $\delta^{15}$N-beta, SP, $\delta^{15}$N-$N_2O$ and $\delta^{18}$O-$N_2O$) would be helpful. The legend says that these are all measurements of USGS51, which should have a SP value of –1.67 at their target of 1000 ppb (1 ppm). However, at that pressure, the SP measured is between –25 and –30‰. Since this is showing "measurement bias," am I to understand that the SP value being measured is between –26.67 and –31.67‰? In addition, what happened to the 5/10/2023 run? The authors do not talk about it in the figure legend or text. Was data from that run discarded? In addition, it would be helpful to show the "experimentally determined, linear correction function" (Line 276) to show how they correct for variation in cell pressure, and how that consistent or not consistent that correction was for all experiments.

**Technical corrections**

There's a little floating "1)" in the upper left corner for Figure 3. Is this supposed to be there? And, the "2" in H2O and CO2 in the figure are not subscripted.

---

## Author Comment (AC1)

| Name | $\delta^{15}N_{AIR}$ | $\delta^{15}N^{\alpha}_{AIR}$ | $\delta^{15}N^{\beta}_{AIR}$ | $\delta^{18}O_{VSMOW\text{-}SLAP}$ | $S_P$ |
|------|------|------|------|------|------|
| USGS51 | $+1.32 \pm 0.04$ ‰ | $+0.48 \pm 0.09$ ‰ | $+2.15 \pm 0.12$ ‰ | $+41.23 \pm 0.04$ ‰ | $-1.67$ ‰ |
| USGS52 | $+0.44 \pm 0.02$ ‰ | $+13.52 \pm 0.04$ ‰ | $-12.64 \pm 0.05$ ‰ | $+40.64 \pm 0.03$ ‰ | $+26.15$ ‰ |

**Figure 1**: Certified values from USGS51 and USGS52 as shown on the manufacturer certificate.

[Figure]

**Figure 2**: Top – left: Pressure effect on $N_2O$ (ppb). Top – Right: Pressure effect on SP-$N_2O$. Bottom: Polynomial fits of the pressure dependence on $N_2O$ or SP-$N_2O$.

---

## Author Comment (AC2)

| Name | $\delta^{15}N_{AIR}$ | $\delta^{15}N^{\alpha}_{AIR}$ | $\delta^{15}N^{\beta}_{AIR}$ | $\delta^{18}O_{VSMOW\text{-}SLAP}$ | $S_P$ |
|---|---|---|---|---|---|
| USGS51 | $+1.32 \pm 0.04$ ‰ | $+0.48 \pm 0.09$ ‰ | $+2.15 \pm 0.12$ ‰ | $+41.23 \pm 0.04$ ‰ | $-1.67$ ‰ |
| USGS52 | $+0.44 \pm 0.02$ ‰ | $+13.52 \pm 0.04$ ‰ | $-12.64 \pm 0.05$ ‰ | $+40.64 \pm 0.03$ ‰ | $+26.15$ ‰ |

**Figure 1**: Certified values from USGS51 and USGS52 as shown on the manufacturer certificate.

[Figure]

**Figure 2**: Top – left: Pressure effect on $N_2O$ (ppb) at four different $N_2O$ mole fractions between 380 and 3300 ppb. Top – Right: Pressure effect on SP-$N_2O$. Bottom: Polynomial fits of the pressure dependence on $N_2O$ or SP-$N_2O$.

| Exp | Culture | n.N2O..nn | g.DW..g.L. | Rates..N2( | X..N.Used | Rep | Species | date.time | sample.ID | d15Na | d15Nb | d18O | d15Nbulk | SP..1.poin | SP..2.poin | U_SP | X | X.1 | d18O_low | d18O_high | d15Nbulk | d15Nbulk |
|---|---|---|---|---|---|---|---|---|---|---|---|---|---|---|---|---|---|---|---|---|---|---|
| 2 | C. reinhard | 690.86 | 1.067 | 359.7105 | 0.276344 | 1 | C. reinhard | 16/11/2023 10:42 | #1_ID-173 | -92.5 | -118.4 | 27.2 | -105.5 | 26 | 25.2 | 1.07 | NA | NA | 33.2 | 34.2 | -106.9 | -133.9 |
| 2 | C. reinhard | 669.34 | 0.99 | 375.6117 | 0.267736 | 2 | C. reinhard | 16/11/2023 17:13 | #4_ID-173 | -92.8 | -119.2 | 27.3 | -106 | 26.4 | 25.6 | 1.14 | NA | NA | 33.3 | 34.3 | -107.4 | -134.4 |
| 2 | C. reinhard | 725.13 | 0.833 | 483.6134 | 0.290052 | 3 | C. reinhard | 16/11/2023 23:43 | #5_ID-173 | -92.3 | -118.8 | 27.1 | -105.6 | 26.5 | 25.7 | 1.11 | NA | NA | 33.1 | 34.1 | -107 | -134 |
| 2 | C. reinhard | 427.73 | 1.1 | 216.0253 | 0.171092 | 4 | C. reinhard | NA | NA | NA | NA | NA | NA | NA | NA | NA | NA | NA | NA | NA | NA | NA |
| 2 | C. vulgaris | 800.47 | 1.15 | 386.7005 | 0.320188 | 1 | C. vulgaris | 07/10/2023 08:25 | #1_ID-173 | -102.1 | -127 | 28.7 | -114.5 | 24.9 | 24.2 | 0.84 | NA | NA | 34.7 | 35.7 | -115.9 | -142.9 |
| 2 | C. vulgaris | 800.47 | 1.15 | 386.7005 | 0.320188 | 1 | C. vulgaris | 14/10/2023 10:58 | #4_ID-173 | -102 | -127.6 | 30.4 | -114.8 | 25.6 | 24.7 | 0.96 | NA | NA | 36.4 | 37.4 | -116.2 | -143.2 |
| 2 | C. vulgaris | 1413.65 | 1.04 | 755.1549 | 0.56546 | 2 | C. vulgaris | 07/10/2023 14:56 | #4_ID-173 | -102.5 | -127.4 | 30.3 | -114.9 | 25 | 24.3 | 0.84 | NA | NA | 36.3 | 37.3 | -116.3 | -143.3 |
| 2 | C. vulgaris | 1629.17 | 1.63 | 555.2727 | 0.651668 | 3 | C. vulgaris | NA | NA | NA | NA | NA | NA | NA | NA | NA | NA | NA | NA | NA | NA | NA |
| 2 | C. vulgaris | 1245.83 | 1.55 | 446.5341 | 0.498332 | 4 | C. vulgaris | 07/10/2023 21:27 | #5_ID-173 | -101.7 | -126.1 | 29.4 | -113.9 | 24.5 | 23.8 | 1.02 | NA | NA | 35.4 | 36.4 | -115.3 | -142.3 |
| 2 | C. vulgaris | 1890.07 | 1.4 | 750.0278 | 0.756028 | 5 | C. vulgaris | NA | NA | NA | NA | NA | NA | NA | NA | NA | NA | NA | NA | NA | NA | NA |
| 2 | M. aerugin | 1683.69 | 1.95 | 479.6838 | 0.673476 | 1 | M. aerugin | 14/10/2023 17:29 | #4_ID-173 | -110.5 | -111.1 | 16.2 | -110.8 | 0.7 | 0.6 | 0.97 | NA | NA | 22.2 | 23.2 | -112.2 | -139.2 |
| 2 | M. aerugin | 1683.69 | 1.95 | 479.6838 | 0.673476 | 1 | M. aerugin | 05/10/2023 11:52 | #1_ID-173 | -117.6 | -113.8 | 8.6 | -115.7 | -3.8 | -3.8 | 1.7 | NA | NA | 14.6 | 15.6 | -117.1 | -144.1 |
| 2 | M. aerugin | 1088.86 | 1.44 | 420.0849 | 0.435544 | 2 | M. aerugin | 06/10/2023 00:54 | #5_ID-173 | -111.7 | -114.3 | 15.3 | -113 | 2.6 | 2.6 | 1.57 | NA | NA | 21.3 | 22.3 | -114.4 | -141.4 |
| 2 | M. aerugin | NA | 0.93 | NA | NA | 3 | M. aerugin | NA | NA | NA | NA | NA | NA | NA | NA | NA | NA | NA | NA | NA | NA | NA |
| 2 | M. aerugin | 600.72 | 0.99 | 337.1044 | 0.240288 | 4 | M. aerugin | 05/10/2023 18:23 | #4_ID-173 | -116.6 | -114.3 | 9.5 | -115.4 | -2.2 | -2.2 | 1.47 | NA | NA | 15.5 | 16.5 | -116.8 | -143.8 |
| 2 | M. aerugin | 842.94 | 1.15 | 407.2174 | 0.337176 | 5 | M. aerugin | NA | NA | NA | NA | NA | NA | NA | NA | NA | NA | NA | NA | NA | NA | NA |
| 1 | Denitrifier | 38276.95 | 7.16 | 2969.968 | 15.31078 | 1 | Denitrifier | NA | NA | NA | NA | NA | NA | NA | NA | NA | NA | NA | NA | NA | NA | |
| 1 | Denitrifier | 30216.08 | 7.72 | 2174.444 | 12.08643 | 2 | Denitrifier | NA | NA | NA | NA | NA | NA | NA | NA | NA | NA | NA | NA | NA | NA | |
| 1 | Denitrifier | 36262.01 | 8.1 | 2487.106 | 14.5048 | 3 | Denitrifier | 14/07/2023 15:24 | #7_17184 | -12.8 | -9.5 | 23 | -11.2 | -3.3 | -3.3 | 0.89 | NA | NA | 29 | 30 | -12.6 | -39.6 |
| 1 | Denitrifier | 36262.01 | 8.1 | 2487.106 | 14.5048 | 3 | Denitrifier | 21/07/2023 12:42 | #5_17184 | -11.6 | -8.5 | 25.6 | -10 | -3.1 | -3.1 | 1.35 | NA | NA | 31.6 | 32.6 | -11.4 | -38.4 |
| 1 | C. reinhard | 510.53 | 0.72 | 393.9275 | 0.204212 | 1 | C. reinhard | 12/07/2023 15:31 | #5_17188 | -92.8 | -115.9 | 23.9 | -104.3 | 25.9 | 26.6 | 1.18 | NA | NA | 29.9 | 30.9 | -105.7 | -132.7 |
| 1 | C. reinhard | 551.14 | 0.8 | 382.7361 | 0.220456 | 2 | C. reinhard | NA | NA | NA | NA | NA | NA | NA | NA | NA | NA | NA | NA | NA | NA | NA |
| 1 | C. vulgaris | 3536.21 | 1.36 | 1444.53 | 1.414484 | 1 | C. vulgaris | NA | NA | NA | NA | NA | NA | NA | NA | NA | NA | NA | NA | NA | NA | NA |
| 1 | C. vulgaris | 3069.8 | 1.46 | 1168.113 | 1.22792 | 2 | C. vulgaris | NA | NA | NA | NA | NA | NA | NA | NA | NA | NA | NA | NA | NA | NA | NA |
| 1 | M. aerugin | 2034.46 | 1.68 | 672.7712 | 0.813784 | 1 | M. aerugin | 12/07/2023 10:08 | #1_17186 | -115.4 | -125.3 | 7.5 | -120.3 | 13 | 13.4 | 1.22 | NA | NA | 13.5 | 14.5 | -121.7 | -148.7 |
| 1 | M. aerugin | 1701.55 | 1.28 | 738.52 | 0.68062 | 2 | M. aerugin | NA | NA | NA | NA | NA | NA | NA | NA | NA | NA | NA | NA | NA | NA | NA |

**Figure 3.** Screenshot of the raw data that will be uploaded separately. Some of the data will also be included in the manuscript.

---

## Author Response (AR1)

We thank the Referees and editor for praising the impact of the work and for their comments.

As can be seen in our revised manuscript, we have considered all comments (**Referees` comments are in bold**) and modified the manuscript accordingly. Our point-by-point response can be found below.

Referee 1:

**Firstly, did the authors perform any kind of abiotic, illuminated control? Recent work has shown that sunlight can drive abiotic photochemical $N_2O$ production (Leon-Palmero et al., 2025), and it seems possible that this was occurring in the authors' experiments.**

We did not include an abiotic illuminated control during this study. We followed an established protocol for the production of $N_2O$ in laboratory microalgae cultures (Guieysse et al., 2013) and as we indicated Li 77, the cultures were kept in darkness. Consequently, it is unlikely that abiotic photochemical $N_2O$ production occurred in our samples. From our previous work (Guieysse et al., 2013; Plouviez et al., 2017) abiotic $N_2O$ production remained always low in our experimental setting.

However, we agree that under real setting (i.e. natural environment) this abiotic production must be taken into consideration. We therefore modified Section 3. Environmental implications (Li 152) to: In natural environments, $N_2O$ can be abiotically produced by chemo-denitrification (Stanton et al., 2018) or photochemically (Lean-Palmero et al., 2025). In addition, $N_2O$ can be both produced and consumed…'

**Secondly, the authors added 10 mM $NaNO_2$ to their cultures, which is orders of magnitude higher than the amount of nitrite in natural aquatic environments. Did the authors do any kind of experiment, feeding the cultures lower levels of nitrite to ascertain if the organisms would still produce $N_2O$ under less nutrient-laden conditions?**

We agree that the concentration of $NaNO_2$ used is significantly higher than what would be expected in natural environments. As our focus was to identify the isotopic signature, we used a protocol known to trigger a strong $N_2O$ production in microalgae and cyanobacteria (Guieysse et al., 2013; Plouviez et al., 2017; Fabisik et al., 2023) in order to facilitate ease of detection (over relevance). It should also be noted that our prior work revealed a linear correlation between $NaNO_2$ concentration (up to 12 mM) and $N_2O$ production in *C. vulgaris*, *C. reinhardtii* and *M. aeruginosa*, with $N_2O$ production being 3-5 fold lower at 3 mM than at 12 mM.

**The authors provide the $N_2O$ site preference produced by each organism, but to incorporate this process into models, it is critical to also know the $\delta(^{15}N^{\alpha})$ and $\delta(^{15}N^{\beta})$ as well. What were the $\delta(^{15}N^{\alpha})$ and $\delta(^{15}N^{\beta})$ of the $N_2O$ produced by each organism, and what was the $\delta(^{15}N)$ of the nitrite that they were supplied? This would allow us to calculate an isotope effect and thus incorporate this process into biogeochemical models.**

We agree that establishing the fractionation factors for nitrogen and oxygen during the reduction of $NO_2^-$ to $N_2O$ would be useful – not just for biogeochemical models but also for elucidating the different biochemical pathways of reduction between eukaryotes and prokaryotes that our $N_2O$ isotope and isotopomer results suggest. A comparison of the

'starting' isotopic enrichment of $NO_2^-$ v the 'product' enrichment reported in $N_2O$ would be a useful first step towards establishing such fractionation factors. However, we did not include this in our manuscript because the laboratory where these experiments were carried out at Massey University was closed and all the reagents used in the experiments thrown out. This is regrettable. However, we note that, because the same salt (with the same isotopic composition) was used across all experiments the uncertainty associated with this calculation will not alter the observed pattern of difference between the organisms nor the conclusions drawn from them.

We used the range of $d^{15}N$ and $d^{18}O$ enrichment reported for $NO_2^-$ salt solutions as possible end-member values to parameterise the potential range of fractionation factors for the different organismal $N_2O$ production pathways reported here. We added these estimates (i.e. $d^{15}N-NO_2$ range of -16 to -61‰, and $d^{18}O-NO_2$ range of +6 to +14‰) to Table 1. And we provided further information in Section 4.6.6:

'Similar to Rohe et al. (2017) and Lwicka-Szczeba et al (2017), bulk isotope values ($\delta^{15}N$-$N_2O$ and $\delta^{18}O-N_2O$) are reported relative to the nitrite substrate ($\delta^{15}N-NO_2^-$) and incubation water ($\delta^{18}O-H_2O$), respectively. During our study $\delta^{18}O-H_2O$ was estimated from local surface water $\delta^{18}O-H_2O$ composition, which ranges from -6 to -7 ‰ (Baisden et al. 2017; Whitehead & Booker 2020; Yang et al 2021). As all experiments were run using the same $NaNO_2$ substrate, the $\delta^{15}N-NO_2^-$ composition was estimated by applying the range of reported denitrifying $NO_2^-$ to $N_2O$ enrichment factors (-12‰ (Wei et al. 2019) to -39‰ (Sutka et al. 2003)) to the $\delta^{15}N-N_2O$ composition measured from bacterial denitrification. This yielded a likely $\delta^{15}N-NO_2^-$ range from +1.4 ‰ to +28.4 ‰. Accordingly, the reported variability in bulk isotope values from our study primarily reflects uncertainty in source values rather than measurement or environmental variability.'

**The authors point to other studies showing how phototrophs produce $N_2O$ from NO within the cell, but the vastly different site preferences of the eukaryotic and prokaryotic $N_2O$ suggest different mechanisms. Could the authors speculate on possible different reaction mechanisms for the two kinds of organisms, even though the intermediate (NO) may be the same?**

The reviewer raised a good point. Proteins with similar functions (nitric oxide reductases) are involved in the reduction of NO into $N_2O$. The eukaryotic and prokaryotic proteins are members of distinct families and consequently are structurally different (Hendriks et al., 2000). This could explain the differences between the cite preferences measured. While we prefer not to speculate as further experimental evidence would be needed, we will clarify that point in the manuscript. As can be seen Section 2.1 (Li 75 – 106) has been modified to discuss the potential biochemical pathways involved and the future research needed.

**Line-by-line comments:**

**Line 157: It seems possible that there may have also been photochemical $N_2O$ production in the authors' experiments.**

As mentioned above, this was unlikely in our experimental design, but we now acknowledge the photochemical production from natural ecosystems.

**Line 174: What is "instrument-grade" $N_2$?**

This refers to a purity level of at least 99.99% $N_2$. This has been clarified in the manuscript (Li 206-207).

**Line 233: What does "indicative" mean in this context?**

The $N_2O$ mole fraction values reported were raw data and only used for sample processing purposes. This has been clarified in the manuscript (li 273).

**Lines 283-284: Include the $\delta(^{15}N^{bulk})$ and $\delta(^{18}O)$ from both gases in Table 3 to illustrate this.**

We have updated Table 3 to include all certified values from USGS51 and USGS52.

**Line 288: How does the uncertainty calculated this way compare to the standard deviation of replicate samples?**

Typical values of the propagated uncertainty scale are around 1.2 ‰ (Table 1). The reproducibility for SP-N2O is around 0.4 ‰, with an accuracy of –0.3 ‰ (Figure 1). The analytical steps for the experiments included in the reproducibility assessment are identical to the analytical steps of the sample analysis for each sample measurement sequence. The propagated uncertainty is therefore a conservative uncertainty estimate.

**Line 290: The term $U_{REF\_span}^2$ should be multiplied by the correction factor, squared.**

We changed the equation Li 341 to:

$$U_{tot} = SQRT(U_{sam}^2 + U_{REF\_a}^2 + U_{REF\_b}^2 + U_{REF\_span}^2 \times F_{SPAN}^2 + U_{p\text{-corr}}^2 + U_{N2O\text{-amount-corr}}^2)$$

where $F_{SPAN}^2$ is the factor of the span correction.

**Line 295: Not the standard error of the slope? Also, it would be highly useful to see a visual representation of these correction functions.**

In addition to a Figure showing the $N_2O$ amount effect on SP-N$_2$O values, we have included an additional Figure for the pressure effect (See Supplementary Information 3). In addition, we have included the following paragraph in the Appendix section (Li 264-268).

'The pressure correction was determined using four gas mixtures with $N_2O$ mole fractions of 380 ppb, 1080 ppb, 2100 ppb and 3300 ppb. The effect of variable cell pressure on $N_2O$ mole fractions and all measured isotope species was linear across the relevant pressure range (Figure S3, Supplementary Information 3). However, the slope of that effect changed with the $N_2O$ mole fraction. Slopes of the pressure corrections for $N_2O$ and all isotopomer species were determined using polynomial fits (Figure S3, Supplementary Information 3).'

**Line 455 and elsewhere: The formatting of the tables is confusing and difficult to read.**

We have reformatted the Tables (e.g. change the orientation to landscape to extend the size of the columns) to improve readability.

**Plouviez et al. take on a significant problem within the biogeosciences – namely, that the N₂O budget is not closed, and one of the main hurdles in closing the budget is accurately accounting for all sources and sinks of N₂O since it can be produced / consumed by multiple biotic and abiotic pathways. A powerful tool in this space is measuring "Site Preference" (SP), which quantifies the relative 'preference' of $^{15}$N for the central ('alpha') or outer ('beta') site in the asymmetrical, linear N₂O molecule. Therefore, many groups have been working to systematically measure the SP of all known sources and sinks of N₂O in an effort to close the N₂O budget, as well as identify sources of N₂O that may be mitigated to prevent greenhouse gas emissions.**

**Plouviez et al. measure the N₂O SP of eukaryotic and bacterial photosynthesizers, which have been shown to produce N₂O outside of the metabolic pathways that N₂O production has been typically attributed to (denitrification and nitrification, either by bacteria or fungi). Measuring this value is particularly important for understanding N₂O cycling in the oceans, since denitrifiers, nitrifiers, and algae all coexist in complex microbial assemblages – therefore, finding potentially unique SP signatures for algae may help disentangle complex marine N₂O cycling. They state that they describe a new method for the accurate laser-based analysis of N₂O isotopes, which enables them to conduct novel SP measurements of algal N₂O. They find significantly different SP signatures for the eukaryotic algae (*C. reinhardtii* and *C. vulgaris*) compared to the cyanobacteria (*M. aeruginosa*).**

We thank Reviewer 2 for acknowledging the impact of our study and for his/her thorough review.

**I have two main comments regarding this manuscript. The first is about the technical aspects of the measurement. The second is about the interpretation of the isotopic signatures.**

**For the first point, I cannot independently evaluate the quality of the data presented because it lacks key outputs that would enable independent calculation of this. I assume that authors are measuring the major isotopologues $^{14}$N$^{14}$N$^{16}$O (446), $^{14}$N$^{15}$N$^{16}$O (456), $^{15}$N$^{14}$N$^{16}$O (546) and $^{14}$N$^{14}$N$^{18}$O (448) and not the rarer clumped species (i.e. $^{14}$N$^{15}$N$^{18}$O), though this is never explicitly stated.**

Our instrument is a Los Gatos Research (LGR, now ABB) analyser. LGR does not manufacture an analyser that measures clumped isotopes. We will clarify that point when describing the equipment in the materials and methods:

"LGR measures only major isotopologues and neither clumped species nor $\delta^{17}$O-N₂O."

**SP is calculated as the relative difference between the $^{15}$N isotopologues (SP = d456 – d546) and the bulk nitrogen isotopic composition is the average of the alpha and beta sites (d$^{15}$N$_{bulk}$ = (d456 – d546)/2; see Kanterova et al. 2022 *GCA*, for example, for calculations) – therefore, understanding issues of sample bracketing, variations among samples, and etc. can be masked by reporting SP only. This is because variations in SP can be driving by variations in one isotopologue alone, since SP simply describes a relative difference in 456 and 546.**

The reviewer correctly states that SP is calculated as d456 – d546. However, SP is not the relative difference between d456 and d546, but the absolute difference between the d456 and d546 values. If we were finding simultaneous changes in both, d456 and d546, and if these changes occurred with the same magnitude and directions for both isotopologues, the effect on SP would be zero. For clarity, we are now presenting more data as further described below.

**In addition, as noted in Griffith 2018 *GMT*, several commercial manufacturers offer optical analyzers based on laser or FTIR spectroscopy that report results in various ways – as an isotopologue mole fraction and/or total mole fractions and/or in 'traditional' isotope delta values. Plouviez et al. do not report the equations used to convert from raw, instrument measurements to final delta values. They also do not report d456 and d546 (also denoted as $d^{15}N$-alpha and $d^{15}N$-beta), nor do they show that the calculated $d^{15}N_{bulk}$ from these values match the measured $d^{15}N_{bulk}$ values. They do report some necessary data in reporting "a new method" – i.e. Figure 1 and 5 – but, again. they do not report their full dataset and only their final calculated values. For example, Table 1 gives the averaged isotopic measurements across all replicates for each species, but the individual measurements behind each average are not in the main text or supplement. Therefore, it is difficult to independently evaluate the quality of their data. I would encourage the authors to publish a more complete dataset, as well as equations involved in converting from raw, instrument measurements to final, reported delta values. This could be amended to the existing supplemental.**

Thank you for highlighting the value of complete data availability. We have uploaded spreadsheets showing the complete outputs from our incubation experiments (including corrected values for each measured isotopologue for each replicate sample and uncertainties). The 'Data availability' statement now includes a link and DOI to this data repository. We are also presenting more data for each isotopologue in Figure 1 and Supplementary Information 3.

**For the second point, I would: 1) Encourage the authors to comment more on the potential mechanism behind the large difference in SP values between the eukaryotic vs. bacterial algal strains; and 2) Have some clarifying questions regarding controls in their experimental systems. As the authors are likely aware of, in both eukaryotic and bacterial algae, it is thought that there are primarily two sources of $N_2O$: flavodiiron proteins (FLV) and cytochrome p450s (CYP55). FLVs are used in pseudo-cyclic electron flow for Photosystem I (PSI) photoprotection, where electrons are put onto $O_2$ instead of being used to generate NADPH. It has been shown that NO can be reduced instead of $O_2$, generating $N_2O$ in the process (Burlacot et al 2020 *PNAS*). CYP55s are a broad class of enzymes involved in multiple metabolic pathways, including pigment biosynthesis and lipid metabolism – i.e., reactions not involved in the light reactions of photosynthesis. Hence, as noted by the authors, it has been shown in *C. reinhardtii* that FLV produce $N_2O$ in the light, while CYP55 produces $N_2O$ in the dark (Burlacot et al. 2020 *PNAS*). Due to similarities between the species, *C. vulgaris* should use a similar pathway, as noted by the authors. Prior work by some of the authors (Fabisik et al. 2023 *Biogeosciences*) performed a BLASTP search on *M. aeruginosa* and found hits for FLV and CYP55, suggesting that similar pathways exist in this strain as well.**

**Plouviez et al. perform all cell suspensions in the dark – this should isolate the CYP55 signal. The SP signals from *C. reinhardtii* and *C. vulgaris* are similar to that of the fungal nitric oxide reductase (Figure 2), which also belongs to the CYP55 family. However, the SP signals from *M. aeruginosa* are quite different and better match the bacterial nitric oxide reductase (Figure 2). One interpretation of their results is that CYP55 from eukaryotic and bacterial algae are quite different, and that is reflected in their $N_2O$ SP values – this appears to be the primary interpretation that the authors make, though they do not attribute it to the enzyme explicitly. Alternatively, in *M. aeruginosa*, since they note that the pathway has not been fully 'elucidated,' non-CYP55 sources of $N_2O$ may be possible. Potentially relevant, an enzyme called flavohemoglobin protein (FHP) has recently been measured for $N_2O$ SP (Wang et al. 2024 *PNAS*). FHP is similar to FLV as they both have flavins as a co-factor – diflavins like FLV have two, while FHP has a flavin and heme cofactor. Measured $N_2O$ SP values in Wang et al. 2024 *PNAS* of FHP from *P. aeruginosa*, *A. baumannii* and *S. aureus* are similar to those measured from *M. aeruginosa* in this paper (roughly 0 to 15‰ in Wang et al., depending on strain, compared to 2±7‰ for *M. aeruginosa* in this paper). The authors should also explicitly note if *M. aueruginosa* has a nitric oxide reductase (NOR) or not, since that would aid in interpretation of this unique signal. In addition, it may be potentially relevant that the standard deviation of their reported SP values from *M. aeruginosa* is much larger than that of *C. reinhardtii* and *C. vulgaris*, though they do not report the non-averaged data nor the 456 / 546 data, so this is difficult to interpret. That may also help better interpret why the SP values of *M. aeruginosa* are so different.**

Nitrous oxide can be synthesized via several biotic and abiotic synthetic pathways. To accurately attribute $N_2O$ sources in complex environments it is essential to consider the full spectrum of biotic and abiotic processes that may contribute to its production.

We have therefore improved section 2.1 to suggest the different $N_2O$ biosynthetic routes in microalgae and cyanobacteria (Li 76-106). We also modified Section 2.3. (Li 119 – 151) to:

'The eukaryotic microalgae (*C. reinhardtii* and *C. vulgaris*) and the cyanobacteria tested synthesized $N_2O$ and consistently produced a SP-$N_2O$ signature demonstrating a clear isotope preference during $N_2O$ production (Table 1). The SP-$N_2O$ signatures of the eukaryotic microalgae were similar (25.8 ± 0.59 ‰ and 24.1 ± 0.37 ‰, respectively) and significantly different to the SP-$N_2O$ from *M. aeruginosa* (2.1 ± 6.8 ‰), meaning this could indeed be used to distinguish between photosynthetic $N_2O$ producers.

With several biochemical pathways potentially involved and unknowns (e.g. which protein is involved in *M. aeruginosa* NO reduction to $N_2O$), consideration is, however, needed. The similarity of the isotopic signatures from the eukaryotic microalgae could be expected considering that both are chlorophyta, confirming that this division uses a consistent $N_2O$ biosynthetic pathway (Bellido-Pedraza et al., 2020; Plouviez et al., 2017). This needs to be further confirmed by testing other chlorophyta and eukaryotic taxons (Timilsina et al., 2022).

The SP-value measured for *M.aeruginosa* is similar to that reported by Wang et al., 2024 for the bacteria *P. aeruginosa*, meaning that *M. aeruginosa* could use a similar biochemical pathway for $N_2O$ synthesis. However, no hits were found from the BLASTP search for the flavohemoglobin (A0A0H2ZC95) or NORb (A0A0H2ZLE2) or NORc (A0A0H2ZKE8) involved in NO reduction to $N_2O$ in *P. aeruginosa* (Wang et al., 2024). While *M. aeruginosa* harbour a homolog to *C. reinhardtii* P450, the difference in SP-value would

suggest that a different protein is involved. As suggested in Section 2.1, further research is therefore needed to confirm the protein that catalyses the reduction of NO to $N_2O$ in cyanobacteria.'

**For this point, did the authors repeat their experiments in the light? Given the established light-dependent nature of $N_2O$ production in *C. reinhardtii*, if FLV does have a different $N_2O$ SP,**

The reviewer raised a good point. While we previously showed that the three species tested can also synthesize $N_2O$ in the light (Guieysse et al., 2013; Plouviez et al., 2017; Fabisik et al., 2023), we have not performed any experiment in the light. We instead used a protocol known to trigger a strong $N_2O$ production in microalgae and cyanobacteria (i.e. cultures in darkness supplied with $NaNO_2$; Guieysse et al., 2013; Plouviez et al., 2017; Fabisik et al., 2023) to facilitate ease of detection (over relevance).

As can be seen above this is now acknowledged in Section 2.1 (Li 91-95):

Because different enzymes are involved according to the light conditions experienced by eukaryotic microalgae (i.e. FLVs vs cytochrome P450), further research is needed to investigate the influence of light on SP-values reported from microalgae.

**In addition, other enzymes besides NOR, FLV, and CYP55 can produce $N_2O$. Currently, nitric oxide reductases (NOR), P450nor, cytochrome P460, cytochrome p450 (CYP55), cytochrome c554, flavodiiron proteins (FLVs) and flavohemoglobin proteins (FHPs) have been shown to produce $N_2O$ as a direct product of an enzymatic reaction (see Ferousi et al. 2020 *Chem Rev*, Kuypers et al 2018 *Nat Rev Microbiol* and Poole & Hughes 2000 *Mol Microbiol* for review). Did the authors attempt to check if their wild-type (WT) strains have genes encoding any of these potential enzymatic sources? This check can be doing through searches like BLASTP, qPCR, RNAseq, or other similar techniques for working on non-genetically tractable strains (i.e. strains where making clean deletions of a certain gene are difficult).**

By performing BLASTP, Bellido-Pedraza et al. (2020) already estimated that nearly one third of the 100 photosynthetic microorganisms described in databases contain homologs of at least one of the proteins involved in $N_2O$ synthesis in *Chlamydomonas* (i.e. NR, NirK, CYP55 and FLVs), including *Chlorella vulgaris* and *Microcystis aeruginosa*.

Noteworthy, no hits were found from the BLASTP search for the flavohemoglobin (A0A0H2ZC95) or NORb (A0A0H2ZLE2) or NORc (A0A0H2ZKE8) involved in NO reduction to $N_2O$ in *P. aeruginosa* (Wang et al., 2024). Consequently, as we have now acknowledged, further research is still needed to identify the protein involved in $N_2O$ synthesis in cyanobacteria (Li 138 – 139).

While beyond the scope of the current study, we have provided more details about the potential pathways involved (Section 2.1), and we acknowledge the need for further research to determine the influence of conditions and species on SP-values.

Considering that different pathway may be involved according to the conditions experienced by phototrophs, we propose to modify the title of our study to:" A novel laser-based spectroscopic method reveals the isotopic signatures of nitrous oxide produced by eukaryotic

and prokaryotic phototrophs in darkness"; and to moderate the aim and impact of our study (end of the introduction, Li 69-72):

"Here we describe a new method for the accurate laser-based analysis of $N_2O$ isotopes, which has enabled us to, for the first time, measure the SP-$N_2O$ signatures of microalgae in darkness. Our study demonstrated that microalgae have specific SP-$N_2O$ signatures. While further research is needed, our study is a first step to ultimately develop process-specific $N_2O$ monitoring from aquatic ecosystems."

**Finally, regarding experimental controls, it is established that $N_2O$ can be produced abiotically (i.e. 'chemodenitrification' Stanton et al. 2018 *Geobiology*), and this process is strongly pH-dependent, where acidic pHs produce nitric oxide radicals that can then be further reduced to N2O (i.e. Su et al 2019 *ES&T*). Did the authors control or check pH of their growth media? Though the media composition is given, there is no indication that the pH of the system was checked prior to incubation, or what the target pH of their media is. In addition, did the authors perform any no-cell controls, where the media was incubated with no cells? I may have missed this, but it does not appear that the authors did this.**

The pH of the cultures was monitored at the beginning and at the end of the experiment and ranged from 6.98 - 7.32 (the media used was buffered). As can be seen from our reply to Reviewer 1, from our previous work (Guieysse et al., 2013; Plouviez et al., 2017) abiotic $N_2O$ production always remained low in our experimental setting. Consequently, we have not performed abiotic controls during this study. Noteworthy, the protocol i.e. cultures supplied with high nitrite under darkness is used to obtain a strong biotic response. Hence the potential issues caused by abiotic $N_2O$ synthesis were minimized by design. We included the following in Section 3 (Li 153 – 154): 'In natural environments, $N_2O$ can be abiotically produced by chemo-denitrification (Stanton et al., 2018) or photochemically (Lean-Palmero et al., 2025). In addition, $N_2O$ can be produced and consumed by organisms….'

Last sentence of Section 3. (Li 164 – 166):

'As mentioned above, $N_2O$ can be synthesized via several biotic and abiotic synthetic pathways under natural conditions. To accurately attribute $N_2O$ sources in complex environments it is essential to consider the full spectrum of biotic and abiotic processes that may contribute to its production.'

**In addition, $N_2O$ can be formed readily from NO radicals, which makes it important to control for all sources of NO radicals, particularly in wild-type (WT) strains. Both bacteria and eukaryotes can create NO through a diverse set of nitric oxide synthases (Forstermann & Sessa 2011 *Eur Heart J*), and these NO radicals can spontaneously react to form $N_2O$ in the absence of oxygen. Did the authors check for these potential NO sources in their strains?**

As indicated by the reviewer, the oxidation of arginine to NO and citrulline by the NO synthase (NOS) is a well-known reaction in mammals and bacteria. While some algae including *C. reinhardtii* harbour NOS homologs (Jeandroz et al., 2016), the $NO_2^-$-independent synthesis of $N_2O$ via conversion of L-arginine by NOS was previously ruled out in *C. vulgaris* and *C. reinhardtii* because $N_2O$ synthesis was not observed in $NO_2^-$-free cultures supplied with l-arginine (Guieysse et al., 2013; Plouviez et al., 2017). The synthesis

of NO via NOS activity was also found to be low in comparison to the reduction of $NO_2^-$ by nitrate reductase in *M. aeruginosa* (Tang et al., 2013).

It is therefore unlikely that the synthesis of NO was a result of NOS activity during our study. We added that sentence in Section 2.1 (Li 84-85): "NO synthesis via NOS synthases has previously been ruled out for both *C. vulgaris* and *C. reinhardtii*."

Regardless we respectfully not the objective of this study was not to elucidate a pathway or demonstrate the potential involvement of radicals, which we do not challenge. Instead, our focus is on demonstrating the different SP responses.

**Overall, Plouviez et al. tackle an important problem in the biogeosciences – constraining the $N_2O$ SP of eukaryotic and bacterial photosynthesizers, which produce $N_2O$ outside of the metabolic pathways that $N_2O$ production has been typically attributed to (denitrification and nitrification, either by bacteria or fungi). Their work offers an important starting point for further, more detailed physiological work that will enable this measurement to be used to disentangle complex microbial communities of denitrifiers, nitrifiers, and photoysnthesizers, helping the community close the $N_2O$ budget and disentangle complex marine N2O cycling.**

 **Specific questions**

**The authors fine extremely depleted $d^{15}N_{bulk}$ values of about –100‰. This is outside of the range of their standards, and also of $N_2O$ in air (ranged from ~9 to 6‰ over the past 300 years; Park et al 2012 Nat Geosci). Did they use a very depleted source of nitrite? The $d^{15}N$ of the nitrite supplied should be included.**

The reviewer is correct in that the $d^{15}N$(bulk) values we found were a long way away from the values covered by our reference gas. As can be seen in our response to Reviewer 1, we did not measure the $d^{15}N$ of the nitrite supplied. We instead use the range of $d^{15}N$ and $d^{18}O$ enrichment reported for $NO_2^-$ salt solutions as possible end-member values to parameterise the potential range of fractionation factors for the different organismal $N_2O$ production pathways reported (**Table 1**). The following explanation was added at the end of Section 4.6.6 (Li 229 – 336):

"Similar to Rohe et al (2017) and Lwicka-Szczeba et al (2017), bulk isotope values ($\delta^{15}N$-$N_2O$ and $\delta^{18}O$-$N_2O$) are reported relative to the nitrite substrate ($\delta^{15}N$-$NO_2^-$) and incubation water ($\delta^{18}O$-$H_2O$), respectively. During our study $\delta^{18}O$-$H_2O$ was estimated from local surface water $\delta^{18}O$-$H_2O$ composition, which ranges from -6 to -7 ‰ (Whitehead & Booker 2020; Baisden et al. 2017; Yang et al 2021). As all experiments were run using the same $NaNO_2$ substrate, the $\delta^{15}N$-$NO_2^-$ composition was estimated by applying the range of reported denitrifying $NO_2^-$ à $N_2O$ enrichment factors (-12‰ (Wei et al. 2019) to -39‰ (Sutka et al. 2003)) to the $\delta^{15}N$-$N_2O$ composition measured from bacterial denitrification. This yielded a likely $\delta^{15}N$-$NO_2^-$ range from +1.4 ‰ to +28.4 ‰. Accordingly, the reported variability in bulk isotope values from our study primarily reflects uncertainty in source values rather than measurement or environmental variability."

**In Table 1, what does "F" mean in the footnotes? At first I thought it meant fraction consumed, but one value of F is 1200.**

F represent the F-statistic computed for ANOVA tests of difference. This has been clarified in the footnote of Table 1.

**For Table 3, what are the $d^{15}N$-alpha and $d^{15}N$-beta values for the standards used?**

The data has been included in Table 3.

**For Figure 1, what are the $d^{15}N$-alpha and $d^{15}N$-beta values, not just the SP values?**

We have revised Figure 1 and included the $d^{15}N$-alpha and $d^{15}N$-beta values.

**In addition, just to clarify, only USGS52 was measured over time, and not USGS51 as well?**

Figure 1 shows results from measurements when USGS52 in air samples were measured as quality control standard. Aliquots of USGS52 in air were filled into a sampling bag and measured and processed as every unknown sample. These specific USGS52 in air measurements were only used for quality control monitoring, and not for calibration or data correction.

Noteworthy, independent measurements of USGS51 and USGS52 in air were used in each measurement sequence to assign isotope values to all samples.

**Related to this, I am slightly confused because Figure 5 shows only USGS51 and not USGS52, and Figure 6 suggests that both reference gases were measured regularly.**

That is correct. That Figure only shows the SP over $N_2O$ for USGS51 in air. The same dilution sequence was measured for USGS52 in air in every measurement sequence. The values for USGS52 in air are not shown, as they do not add any additional information.

**For Figure 2a, Wang et al. 2024 *PNAS* offers a more recent compilation of $N_2O$ SP measurements than Denk et al. 2017.**

The value presented in Wang et al., 2024 https://doi.org/10.1073/pnas.2319960121 are in the same order of magnitude as the one presented by Denk et al., 2017. As we are now citing Wang et al., 2024, we have kept the compilation of SP-values from Denk et al., 2017.

**For Figure 2b, the authors are comparing their data vs. that from published denitfier data. In the text (line 113) and in the figure legend, which experimental denitrifer data are the authors comparing their data to? (In addition, the plot should specific 'bacterial denitrifiers' instead of just 'denitrifiers'). Multiple groups have measured bacterial denitrifiers and there is a larger range of values than they show in their figure. For example, see Wang et al. 2024 *PNAS* or Toyoda et al 2017 *Mass Spectrom Rev* for recent compilations. In addition, unless the authors are using nitrite with the exact same $d^{15}N$ as that study, one would not expect the $d^{15}N$-$N_2O$ and $d^{18}O$-$N_2O$ to be the same. Instead, the relative fractionation (i.e. $^{15}e$ or $^{18}e$) are comparable, not the bulk values. Therefore, the epsilons should be calculated and plotted instead.**

We thank the reviewer for his/her suggestions and for motivating us to improve the presentation of our bulk isotope data. It is worth noting that Fig. 2b only showed primary data

produced during this study, hence the relatively small range shown for bacterial denitrification. Besides, as all the cultures were fed the same substrate, the bulk isotope comparison between the species is valid. We have made the following changes:

1. We updated the figure caption to clarify our data v literature data: "(b) Findings from this study on the 3D $N_2O$ isotope composition for microalgae (*C. vulgaris*, *C. reinhardtii*), cyanobacteria (*M. aeruginosa*) and bacterial denitrifier samples, where bulk isotopes ($\delta^{18}O$-$N_2O$ and $\delta^{15}N$-$N_2O$) are reported relative to substrate $H_2O$ ($\delta^{18}O$-$N_2O – \delta^{18}O$-$H_2O$) and $NO_2^-$ ($\delta^{15}N$-$N_2O – \delta^{15}N$-$NO_2^-$), respectively."

2. We updated Fig. 2b legends to specify 'bacterial denitrifiers'

3. We updated the presentation of bulk isotope data so that the values are more universally relevant by applying a correction for the substrate isotope composition (as described in our response to the previous comment).

**For Figure 5, this is something where showing the full suite of data ($d^{15}N$-alpha, $d^{15}N$-beta, SP, $d^{15}N$-$N_2O$ and $d^{18}O$-$N_2O$) would be helpful.**

As can be seen in Figure 1 and Figure S2, these values have been included in the revised manuscript.

**The legend says that these are all measurements of USGS51, which should have a SP value of –1.67 at their target of 1000 ppb (1 ppm). However, at that pressure, the SP measured is between –25 and –30‰. Since this is showing "measurement bias," am I to understand that the SP value being measured is between –26.67 and –31.67‰?**

The values in that Figure are raw data after pressure correction for comparability. We do not expect that these values accurately reflect the certified USGS51 value at any given $N_2O$ amount. This is to be expected because before the "true" value can be determined the instrument response to a specific sample under specific analytical conditions need to be accounted for first (Harris et al., 2020). Reported SP value for USGS51 at $N_2O$ levels around 1 ppm varied between –25 and –45 ‰ in our experiments.

**In addition, what happened to the 5/10/2023 run? The authors do not talk about it in the figure legend or text. Was data from that run discarded?**

We cannot explain why the $N_2O$ amount function appears so different on that day. The sample amount varies with time, which has been observed by other researchers as well. Therefore, the $N_2O$ amount correction function needs to be defined in every measurement sequence.

**In addition, it would be helpful to show the "experimentally determined, linear correction function" (Line 276) to show how they correct for variation in cell pressure, and how that consistent or not consistent that correction was for all experiments.**

The $N_2O$ and isotopomers measurements bias due to $N_2O$ amount and cell pressure dependence are shown in the Supplementary Information 3.

**Technical corrections**

**There's a little floating "1)" in the upper left corner for Figure 3. Is this supposed to be there? And, the "2" in H2O and CO2 in the figure are not subscripted.**

These have been corrected.

**References:**

Bellido-Pedraza, C. M., Calatrava, V., Sanz-Luque, E., Tejada-Jimenez, M., Llamas, A., Plouviez, M., Guieysse, B., Fernandez, E., and Galvan, A.: *Chlamydomonas reinhardtii*, an Algal Model in the Nitrogen Cycle, Plants (Basel), 9, 10.3390/plants9070903, 2020.

Burlacot, A., Richaud, P., Gosset, A., Li-Beisson, Y., and Peltier, G.: Algal photosynthesis converts nitric oxide into nitrous oxide, Proc Natl Acad Sci U S A, 117, 2704-2709, 10.1073/pnas.1915276117, 2020.

Denk, T. R. A., Mohn, J., Decock, C., Lewicka-Szczebak, D., Harris, E., Butterbach-Bahl, K., Kiese, R., and Wolf, B.: The nitrogen cycle: A review of isotope effects and isotope modeling approaches, Soil Biology and Biochemistry, 105, 121-137, https://doi.org/10.1016/j.soilbio.2016.11.015, 2017.

Fabisik, F., Guieysse, B., Procter, J., and Plouviez, M.: Nitrous oxide ($N_2O$) synthesis by the freshwater cyanobacterium *Microcystis aeruginosa*, Biogeosciences, 20, 687-693, 10.5194/bg-20-687-2023, 2023.

Guieysse, B., Plouviez, M., Coilhac, M., and Cazali, L.: Nitrous Oxide ($N_2O$) production in axenic *Chlorella vulgaris* microalgae cultures: evidence, putative pathways, and potential environmental impacts, Biogeosciences, 10, 6737-6746, 10.5194/bg-10-6737-2013, 2013.

Hendriks, J., Oubrie, A., Castresana, J., Urbani, A., Gemeinhardt, S., Saraste, M.: Nitric oxide reductases in bacteria. Biochim Biophys Acta – Bioenerg, 1459, 266-273, 2000.

Jeandroz, S., Wipf, D., Stuehr, D. J., Lamattina, L., Melkonian, M., Tian, Z., Zhu, Y., Carpenter, E. J., Wong, G. K-S., Wendehenne, D. Occurrence, structure, and evolution of nitric oxide synthase–like proteins in the plant kingdom.*Sci. Signal.***9**,re2-re2(2016).

Leon-Palmero, E., Morales-Baquero, R., Thamdrup, B., Löscher, C., and Reche, I.: Sunlight drives the abiotic formation of nitrous oxide in fresh and marine waters, Science, 387, 1198–1203, https://doi.org/10.1126/science.adq0302, 2025.

Lewicka-Szczebak, D., Augustin, J., Giesemann, A., & Well, R. (2017). Quantifying $N_2O$ reduction to $N_2$ based on $N_2O$ isotopocules - validation with independent methods (helium incubation and $^{15}N$ gas flux method). *Biogeosciences, 14*(3), 22. doi:10.5194/bg-14-711-2017

Nitrous oxide from chemodenitrification: A possible missing link in the Proterozoic greenhouse and the evolution of aerobic respiration. *Geobiology*. 2018; 16: 597–609.

Plouviez, M., Wheeler, D., Shilton, A., Packer, M. A., McLenachan, P. A., Sanz-Luque, E., Ocana-Calahorro, F., Fernandez, E., and Guieysse, B.: The biosynthesis of nitrous oxide in the green alga *Chlamydomonas reinhardtii*, Plant J, 91, 45-56, 10.1111/tpj.13544, 2017.

Rohe, L., Well, R., & Lewicka-Szczebak, D. Use of oxygen isotopes to differentiate between nitrous oxide produced by fungi or bacteria during denitrification. *Rapid Communications in Mass Spectrometry, 31*(16), 1297-1312. doi:10.1002/rcm.7909, 2017.

Stanton, C.L., Reinhard, C.T., Kasting, J.F., Ostrom, N.E., Haslun, J.A., Lyons, T.W., Glass, J.B.,

Sutka, R. L., Ostrom, N. E., Ostrom, P. H., Gandhi, H., & Breznak, J. A. Nitrogen isotopomer site preference of N2O produced by Nitrosomonas europaea and Methylococcus capsulatus Bath. *Rapid Communications in Mass Spectrometry, 17*(7), 738-745. doi:10.1002/rcm.968, 2003.

Tang, X., Chen, J., Wang, W-H., Liu, T-W., Zhang, J., Gao, Y-H., Pei, Z-M., Zheng, H-L., The changes of nitric oxide production during the growth of Microcystis aerugrinosa, Environ Poll, 159, 12, 3784-3792, 2011.

Wang, R.Z., Lonergan, Z.R., Wilbert, S.A., Eiler, J.M., Newman, D.K. Widespread detoxifying NO reductases impart a distinct isotopic fingerprint on $N_2O$ under anoxia, Proc. Natl. Acad. Sci. U.S.A. 121 (25) e2319960121, https://doi.org/10.1073/pnas.2319960121, 2024.

Wei, J., Ibraim, E., Brüggemann, N., Vereecken, H., & Mohn, J. First real-time isotopic characterisation of $N_2O$ from chemodenitrification. *Geochimica et Cosmochimica Acta, 267*, 17-32. doi:https://doi.org/10.1016/j.gca.2019.09.018, 2019.